# PARETO POLICY POOL FOR MODEL-BASED OFFLINE REINFORCEMENT LEARNING

**Yijun Yang**[1,4]**, Jing Jiang**[1]**, Tianyi Zhou**[2,3]**, Jie Ma**[1]**, Yuhui Shi**[4]

[1]Australian Artificial Intelligence Institute, University of Technology Sydney
[2]University of Washington, Seattle, [3]University of Maryland, College Park
[4]Department of Computer Science and Engineering, Southern University of Science and Technology
{yijun.yang-1, jie.ma-5}@student.uts.edu.au, jing.jiang@uts.edu.au, tianyizh@uw.edu, shiyh@sustech.edu.cn

## ABSTRACT

Online reinforcement learning (RL) can suffer from poor exploration, sparse reward, insufficient data, and overhead caused by inefficient interactions between an immature policy and a complicated environment. Model-based offline RL instead trains an environment model using a dataset of pre-collected experiences so online RL methods can learn in an offline manner by solely interacting with the model. However, the uncertainty and accuracy of the environment model can drastically vary across different state-action pairs, so the RL agent may achieve a high model return but perform poorly in the true environment. Unlike previous works that need to carefully tune the trade-off between the model return and uncertainty in a single objective, we study a bi-objective formulation for model-based offline RL that aims at producing a pool of diverse policies on the Pareto front performing different levels of trade-offs, which provides the flexibility to select the best policy for each realistic environment from the pool. Our method, "Pareto policy pool (P3)", does not need to tune the trade-off weight but can produce policies allocated at different regions of the Pareto front. For this purpose, we develop an efficient algorithm that solves multiple bi-objective optimization problems with distinct constraints defined by reference vectors targeting diverse regions of the Pareto front. We theoretically prove that our algorithm can converge to the targeted regions. In order to obtain more Pareto optimal policies without linearly increasing the cost, we leverage the achieved policies as initialization to find more Pareto optimal policies in their neighborhoods. On the D4RL benchmark for offline RL, P3 substantially outperforms several recent baseline methods over multiple tasks, especially when the quality of pre-collected experiences is low.

## 1 INTRODUCTION

Offline reinforcement learning (offline RL) (Levine et al., 2020) or batch RL (Lange et al., 2012) can train an agent without interacting with the environment by instead using pre-collected experiences on other agents/policies. Recently, offline RL has been attracting growing interest due to the availability of large-scale datasets of diverse experiences in many RL applications, e.g., autonomous driving (Shin & Kim, 2019), healthcare (Yu et al., 2019), robot control (Schulman et al., 2016), etc. However, RL algorithms developed for the online/interactive setting usually perform poorly in the offline setting (Fujimoto et al., 2019; Janner et al., 2019) due to the data distribution shift caused by (1) the difference between the policy-in-training and the behavior policies used to collect the data; and (2) the difference between the realistic environment in which we will deploy the policy and the environments used to collect the data. These differences can result in function approximation error and biased policy learning (Levine et al., 2020). To address these challenges, model-based RL approaches (Yu et al., 2020; Kidambi et al., 2020; Rafailov et al., 2021; Yu et al., 2021) firstly learn an environment model from a dataset of logged experiences using supervised learning and then conduct online RL interacting with the model. The learned environment model fully exploits the pre-collected experiences and can avoid/reduce the costly interactions with the realistic environment required by RL, hence improving the sample efficiency.

That being said, due to the large size of state-action space, model-based offline RL approaches can still suffer from "model exploitation" (Levine et al., 2020): when the dataset does not contain

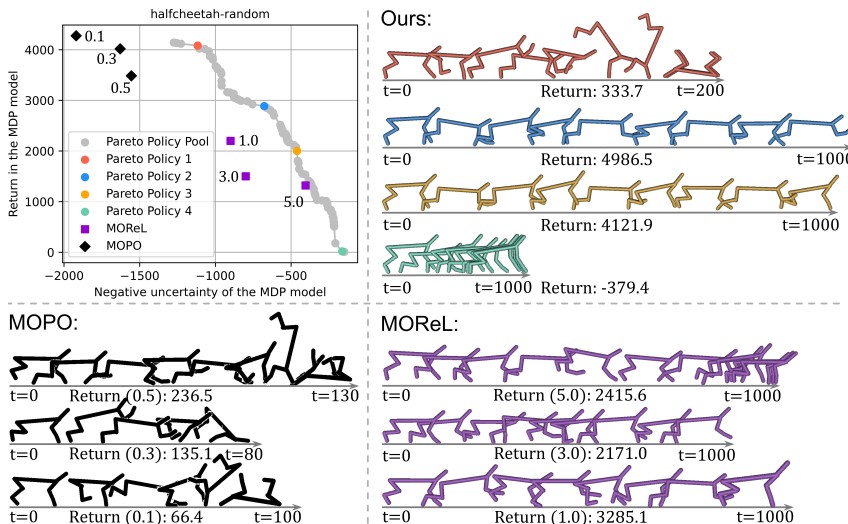

Figure 1: **Model return-uncertainty trade-off** of P3 (ours), MOPO (Yu et al., 2020), and MOReL (Kidambi et al., 2020) on an offline RL dataset "halfcheetah-random" from the D4RL benchmark (Fu et al., 2020). P3 achieves policies with different levels of model return-uncertainty trade-off. MOPO and MOReL are run three times with regularization weights $\{0.1, 0.3, 0.5\}$ and $\{1.0, 3.0, 5.0\}$, respectively. Detailed discussion in Sec. 1.

sufficient samples for some state-action pairs, the epistemic uncertainty of the model on these "out-of-distribution" pairs can lead to a poor policy in offline RL. For example, an RL agent may easily exploit the model by repeatedly visiting the out-of-distribution states where the model erroneously issues higher rewards than the true environment dynamics, and thus the RL objective is biased by overly optimistic evaluation of the agent's performance. A widely-studied strategy to tackle this problem is applying regularization to the reward function, which keeps the model uncertainty small when maximizing the reward issued by the model (Yu et al., 2020; Kidambi et al., 2020; Rafailov et al., 2021). By carefully tuning the regularization weight, it can achieve preferable trade-off between the model reward and model uncertainty. However, it is usually challenging to tune the weight without access to the realistic environment that the learned policy will be deployed to. Moreover, even with access to the realistic environment, hyperparameter tuning methods such as Bayesian optimization (Frazier, 2018) need to run multiple instances of offline RL and can be computationally expensive. In addition, the offline-trained policy may not be directly generalized to different environments. In this paper, we address a challenging problem, i.e., how to balance between the model reward and model uncertainty during offline RL so it can produce diverse policies that can be adapted to different realistic environments?

Instead of formulating the problem as a single-objective optimization with regularization as in previous works, a primary contribution of this paper is to treat the model reward/return and model uncertainty as two separate objectives and develop an efficient bi-objective optimization method producing a pool of diverse policies on the Pareto front, which correspond to different trade-offs between the two objectives. Therefore, when deployed to a new realistic environment, we can accordingly choose the best policy from the pool. Our method is called "Pareto policy pool (P3)," and an example of the policies achieved by P3 is shown in Fig. 1. When deployed to the realistic environment, Pareto policy 1 is overly optimal on the model return, so it runs fast at the beginning but quickly falls to the ground due to the aforementioned "model exploitation". On the contrary, Pareto policy 4 favoring small model uncertainty is overly conservative and keeps standing still for 1000 time-steps because it avoids taking exploratory actions that potentially increase the uncertainty. As expected, Pareto policy 2&3 with the more balanced trade-off between the model return and uncertainty perform better and achieve higher returns in the test environment. These results imply that model-based offline RL's performance significantly relies on the trade-off between model return and uncertainty in the optimization objectives. However, it is usually challenging or intractable to determine the optimal trade-off before deployment and control it during training. Moreover, for existing methods adopting a regularized single-objective, i.e., scalarization (Boyd & Vandenberghe, 2004, Chapter 4.7), even trying all possible regularization weights cannot fully recover the Pareto front and thus cannot guarantee to find the optimal trade-off. For example, in Fig. 1, by running multiple instances with different regularization weights, MOPO (Yu et al., 2020) and MOReL (Kidambi et al.,

2020) can only generate a few separated solutions, and it is difficult to find one with advantageous trade-off among them. In contrast, our method aims at efficiently generating a rich pool of diverse and representative policies covering the entire Pareto front without tuning the trade-off during training. Thereby, when deployed to a new realistic environment, the best policy for the new environment can be efficiently selected from the pool. Hence, P3 provides a simple and principal approach that addresses the two major challenges in model-based offline RL, i.e., "model exploitation" and generalization to different unseen states in order to achieve high returns.

Due to the complicated shape of the Pareto front that is unknown during training, finding a pool of diverse policies covering the whole Pareto front raises several non-trivial algorithmic challenges: (1) How to find policies located at different regions of the Pareto front associated with different levels of model return-uncertainty trade-off? (2) How to avoid training each policy from scratch and reduce the computational cost linearly increasing with the number of policies? Inspired by recent works in multi-objective optimization (Cheng et al., 2016; Ma et al., 2020; Xu et al., 2020), we explore different regions of the Pareto front by generating multiple diverse reference vectors in the bi-objective space, each defining a constrained bi-objective optimization whose solution resides in a local region of the Pareto front. By solving these constrained bi-objective optimization problems, we can obtain a diverse set of policies covering the whole Pareto front. For solving each problem, we extend MGDA algorithm (Désidéri, 2012) to be a two-stage gradient-based method that provably converges to the Pareto front region targeted by the reference vector. In order to achieve more policies on the Pareto front without linearly increasing the cost, we start from the previously obtained policies for initialization and explore their neighborhoods on the Pareto front by perturbing their corresponding reference vectors, resulting in a dense set of Pareto policies in each local region.

In experiments, we evaluate P3 and compare it with several state-of-the-art model-based/free offline RL methods on the standard D4RL Gym benchmark (Fu et al., 2020). P3 achieves the highest average-score over all datasets and significantly outperforms the baseline methods in 5 out of the 9 low/medium-quality datasets, showing the advantages of P3 on learning from non-expert experiences. We also present a thorough ablation study to identify the most important components in P3.

## 2 RELATED WORK

Due to lack of space, we focus our discussion here on directly related works and present a more detailed overview of related work in Appendix A.2. To address the aforementioned "model exploitation", recent works rely on applying uncertainty regularization to the model return (Yu et al., 2020; Kidambi et al., 2020; Rafailov et al., 2021) which can be difficult and costly to tune the weight of regularization. In contrast, our work reframes the policy learning under the environment model as a bi-objective optimization problem and produces a diverse set of policies on the Pareto front performing different levels of model return-uncertainty trade-off, which provides flexibility to select the best policy in the inference stage.

## 3 PRELIMINARIES

We consider an episodic Markov Decision Process (MDP) $\mathcal{M} = \{\mathcal{S}, \mathcal{A}, P, r, H, \rho_0\}$, where $\mathcal{S}$ is the state space, $\mathcal{A}$ is the space of actions, $P$ is the transition probability: $\mathcal{S} \times \mathcal{A} \rightarrow \Delta(\mathcal{S})$ where $\Delta(\mathcal{S})$ is a probability distribution over $\mathcal{S}$, $r : \mathcal{S} \times \mathcal{A} \rightarrow \mathbb{R}$ is the reward function so $r(s_h, a_h)$ is the immediate reward for taking action $a_h$ at state $s_h$, $H$ is the horizon of the process, and $\rho_0$ is the distribution of the initial state $s_0$. RL aims at learning a policy $\pi : \mathcal{S} \rightarrow \mathcal{A}$ maximizing the expected return in Eq. (1), where $\pi$ in this paper takes a deterministic action $a_h$ based on the state $s_h$.

$$\max_{\pi_\theta} R_{\rho_0}(\pi_\theta, \mathcal{M}) = \mathbb{E}_{s_0 \sim \rho_0, \pi_\theta} \left[ \sum_{h=0}^{H-1} r(s_h, a_h) \right]. \tag{1}$$

In model-based offline RL, the agent instead interacts with an environment model rather than the realistic one. We train an environment model $\widehat{\mathcal{M}} = \{\hat{\mathcal{S}}, \mathcal{A}, \hat{P}, \hat{r}, H, \hat{\rho}_0\}$ using a pre-collected dataset $\mathcal{D} \triangleq \{(s_h, a_h, s_{h+1}, r_h) | \pi_b\}$ of experiences by behavior policies, hand-designed controllers, or human demonstrators. By interacting with the model, online RL methods can learn in an offline manner. However, when $\mathcal{D}$ does not contain sufficient samples for some state-action pairs, the epistemic uncertainty of the model on these "out-of-distribution (OOD)" pairs can result in a poor policy. For example, an RL agent may easily exploit the model by repeatedly visiting the OOD states where the model erroneously issues $\hat{r}$ higher than the true environment reward $r$ and thus the RL objective is biased by overly optimistic evaluation of the agent's performance. A widely-studied

strategy to tackle this problem is applying regularization to $\hat{r}$, which keeps the model uncertainty small while maximizing the model reward (Yu et al., 2020; Kidambi et al., 2020; Rafailov et al., 2021). A practical implementation of the regularized reward function is developed by (Yu et al., 2020): $\tilde{r}_h = \hat{r}_h - \lambda u(\hat{s}_h, a_h)$, where $u(\hat{s}_h, a_h)$ denotes the estimation of the model uncertainty at the state-action pair $(\hat{s}_h, a_h)$ and $\lambda$ controls the trade-off between $\hat{r}$ and $u$, which has to be carefully tuned in practice. Based on the regularized reward function, Yu et al. (2020) proposed a modified policy optimization objective:

$$\max_{\pi} \tilde{R}_{\hat{\rho}_0}(\pi, \widehat{\mathcal{M}}) = \mathbb{E}_{s_0 \sim \hat{\rho}_0, \pi} \left[ \sum_{h=0}^{H-1} \left( \hat{r}(\hat{s}_h, a_h) - \lambda u(\hat{s}_h, a_h) \right) \right]. \tag{2}$$

Despite being intuitive, this method's performance is sensitive to the regularization weight $\lambda$ (Kidambi et al., 2020; Yu et al., 2021), and tuning $\lambda$ is usually challenging without access to the realistic environment that the learned policy will be deployed to. Moreover, even granted the access, hyperparameter tuning methods such as Bayesian optimization (Frazier, 2018) require running many instances of offline RL, which can be computationally prohibitive.

Instead of optimizing a single regularized objective, our "Pareto policy pool (P3)" proposed later treats $\hat{r}$ and $u$ as two separate objectives and develop an efficient bi-objective optimization method that does not need to tune the trade-off deliberately but produces a pool of diverse policies, which are learned under different trade-offs between the two objectives. Thereby, when deployed to a realistic environment, the best policy can be chosen from the pool.

## 4 Pareto Policy Pool for Model-based Offline RL

### 4.1 Problem Formulation

In order to estimate the model uncertainty accurately and alleviate the model exploitation problem, we follow previous works (Janner et al., 2019; Yu et al., 2020; Kidambi et al., 2020; Rafailov et al., 2021; Yu et al., 2021) and construct a bootstrap ensemble of $K$ environment models $\{\widehat{\mathcal{M}}^i\}_{i=1}^{K}$. Each model $\widehat{\mathcal{M}}^i = \{\hat{\mathcal{S}}, \mathcal{A}, \hat{P}^i, \hat{r}^i, H, \hat{\rho}_0\}$ is a two-head feed-forward neural network that takes a state-action pair $(\hat{s}_h, a_h)$ as input and outputs the mean $\boldsymbol{\mu}_i$ and standard deviation $\boldsymbol{\sigma}_i$ of $[\hat{s}_{h+1}, \hat{r}_h]$, i.e., the next state concatenated with the reward. More details about our model are given in Appendix A.5. As demonstrated in Yu et al. (2020), this ensemble is effective in estimating the model uncertainty as the maximal standard deviation over all models, i.e., $u(\hat{s}_h, a_h) = \max_{i \in [K]} \|\boldsymbol{\sigma}_i(\hat{s}_h, a_h)\|_2$. Moreover, by randomly selecting a model in each step to provide the reward $\hat{r}(\hat{s}_h, a_h) = \hat{r}_h$, we can effectively mitigate the model exploitation problem. Unlike previous works combining $\hat{r}$ and $u$ as a single objective, we treat them as two separate objectives and aims at solving the bi-objective optimization below.

$$\max_{\theta} \mathbf{J}_{\hat{\rho}_0}(\pi_\theta, \widehat{\mathcal{M}}) = \max_{\theta} (J_{\hat{\rho}_0}^{\hat{r}}(\pi_\theta, \widehat{\mathcal{M}}), J_{\hat{\rho}_0}^{u}(\pi_\theta, \widehat{\mathcal{M}}))^\mathsf{T}, \tag{3}$$

$$(J_{\hat{\rho}_0}^{\hat{r}}(\pi_\theta, \widehat{\mathcal{M}}), J_{\hat{\rho}_0}^{u}(\pi_\theta, \widehat{\mathcal{M}}))^\mathsf{T} = \mathbb{E}_{s_0 \sim \hat{\rho}_0, \pi} \left[ \sum_{h=0}^{H-1} \left( \hat{r}(\hat{s}_h, a_h), \exp\left(-u(\hat{s}_h, a_h)/\kappa\right) \right)^\mathsf{T} \right], \tag{4}$$

where $\kappa$ is a *temperature* applied to $u(\hat{s}_h, a_h)$. For simplicity, in the rest of this paper, we remove $\hat{\rho}_0$ and $\widehat{\mathcal{M}}$. In Eq. (3), $J^{\hat{r}}$ aims to maximize the expected model return, and $J^u$ is designed to minimize the expected cumulative model uncertainty. However, the bi-objective optimization naturally has multiple (up to infinite) optimal solutions instead of one single policy and each optimal policy performs a different level of trade-off between the two objectives. For example, a policy favoring small model uncertainty maybe overly conservative and avoids taking exploratory actions so its model return can be low. In contrast, a policy pursuing high model return might fail in realistic environments at a state that the model is highly uncertain about. Examples of these policies are given in Fig. 1. Formally, for any two policies $\pi_i$ and $\pi_j$, $\pi_i$ dominates $\pi_j$ if and only if $\mathbf{J}(\pi_i) \geq \mathbf{J}(\pi_j)$ and $\mathbf{J}(\pi_i) \neq \mathbf{J}(\pi_j)$. A policy $\pi^*$ is Pareto optimal if no any policy dominates $\pi^*$ in $\mathbb{R}^d$, i.e., no objective can be improved without detriment to another objective at $\pi^*$. All Pareto optimal policies constitute the Pareto set, and the Pareto front is the image of the Pareto set in the space spanned by all the objectives. Unfortunately, it is almost infeasible in practice to find the whole Pareto set and the Pareto front's shape is also unknown.

As discussed in Sec. 3, it is difficult to determine the optimal trade-off between $J^{\hat{r}}$ and $J^u$ because we cannot access the realistic environment during training. It is also challenging and costly to control

the trade-off. Hence, a straightforward strategy is to find a diverse set of policies on the Pareto front performing different levels of trade-off and select the best one when deployed to a realistic environment. However, how to find these Pareto optimal policies is still an open challenge. In addition, it is expensive to train each Pareto policy from scratch. After allocating a few diverse policies on the Pareto front, can we start from them to find more Pareto policies so we can avoid linearly increasing the computational costs? To overcome these challenges, we develop "Pareto policy pool (P3)", which can efficiently and precisely find a diverse set of policies on the Pareto front of return-uncertainty bi-objective optimization.

## 4.2 PARETO POLICY POOL

Fig. 2 illustrates the main idea of "Pareto policy pool (P3)", whose detailed procedure are given in Alg. 1. In order to find diverse Pareto policies and precisely control the trade-off of each policy, P3 generates multiple reference vectors $\{\boldsymbol{v}_i\}_{i=1}^n$ in the objective space, each forming a constraint to the bi-objective optimization in Eq. (3) and targeting a different region on the Pareto front (Sec. 4.2.1). Thereby, the Pareto policies $\{\pi_i\}_{i=1}^n$ achieved by solving the $n$ constrained bi-objective optimizations are diverse in terms of the objective trade-off. For solving each of them, we develop an efficient two-stage optimization algorithm, i.e., Alg. 2, which starts from an initial point and moves to a region fulfilling the constraint (correction stage), and then apply a multi-objective optimization algorithm to find a Pareto policy in the

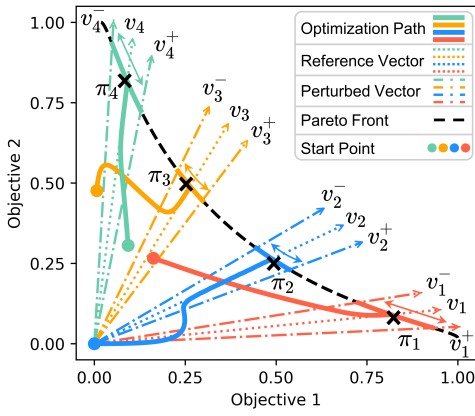

Figure 2: Illustration of P3 by solving a benchmark problem from (Lin et al., 2019).

targeted region. To avoid training each Pareto policy from scratch, which leads to linearly increasing costs for $\geq n$ policies, we develop a local extension method on the Pareto front in Sec. 4.2.2. This "Pareto Extension" firstly generates more reference vectors by perturbing each vector in $\{\boldsymbol{v}_i\}_{i=1}^n$ and then uses their associated Pareto policies $\{\pi_i\}_{i=1}^n$ to warm start the bi-objective optimization constrained by the new vectors (Line 9-13 in Alg. 1). These locally extended policies, together with the $n$ policies, compose a pool of diverse policies on the Pareto front, from which we can select the best policy when deploying them to realistic environments.

---

**Algorithm 1** Pareto policy pool (P3) for model-based offline RL

---

1: **input:** dataset $D$, constraint $\psi < 0$, step size $\eta$, num. reference vectors $n$, $T_g \gg T_l$
2: **initialize:** environment models, Pareto policy pool $\mathcal{P} = \varnothing$, $0 < \tau_a < \tau_b < 1$ for Eq. (5), $0 < \epsilon < \tau_a$ for Eq. (8), number of updates: $T = n(T_g + 2T_l)$
3: Train the model on $D$ using supervised learning;
4: Generate $n$ reference vectors $\{\boldsymbol{v}_1, \ldots, \boldsymbol{v}_n\}$ by Eq. (5);
5: **for** $i \in \{1, \ldots, n\}$ (in parallel) **do**                                                          ▷ Diverse Pareto Policies
6:     Initialize a policy $\pi_i$
7:     **for** $j = 0, 1, \ldots, T_g - 1$ **do**
8:         Update the parameters of $\pi_i$ by Alg. 2 with $\boldsymbol{v}_i$;
9:     Generate $\{\boldsymbol{v}_i^+, \boldsymbol{v}_i^-\}$ to $\boldsymbol{v}_i$ by Eq. (8);                              ▷ Local Pareto Extension
10:     **for** $\boldsymbol{v}' \in \{\boldsymbol{v}_i^+, \boldsymbol{v}_i^-\}$ **do**
11:         **for** $j' = 0, 1, \ldots, T_l - 1$ **do**
12:             $\mathcal{P} = \mathcal{P} \cup \{\pi_i\}$;                                                          ▷ Store Pareto policies into the pool
13:             Update the parameters of $\pi_i$ by Alg. 2 with $\boldsymbol{v}'$;
14: **output:** $\mathcal{P}$;

---

### 4.2.1 DIVERSE PARETO POLICIES BY REFERENCE VECTORS

Inspired by recent works in multi-objective optimization (Cheng et al., 2016; Lin et al., 2019; Ma et al., 2020; Xu et al., 2020), we explore the Pareto front by generating a diverse set of reference vectors defining multiple constrained bi-objective optimization problems. As shown in Fig. 2, we generate $n$

uniformly distributed reference vectors $\{\boldsymbol{v}_i\}_{i=1}^n$ in a 0-1 normalized objective space by Eq. (5), i.e.,

$$\boldsymbol{v}_i \triangleq (\tau_b - (i-1)\tau_c, \tau_a + (i-1)\tau_c), \tau_c = \frac{\tau_b - \tau_a}{n-1}, 0 < \tau_a < \tau_b < 1, i \in \{1, 2, \ldots, n\}, \quad (5)$$

where $\tau_a$ and $\tau_b$ control the range covered by reference vectors. Each $\boldsymbol{v}_i = (v_i^1, v_i^2)$ defines a constrained bi-objective optimization problem based on Eq. (3) whose solution resides in the targeted region of the Pareto front:

$$\max_\theta \mathbf{J}(\pi_\theta) \triangleq \max_\theta (J^{\hat{r}}(\pi_\theta), J^u(\pi_\theta))^\mathsf{T},$$

$$\text{s.t. } \Psi(\pi_\theta, \boldsymbol{v}_i) \triangleq -D_{\mathrm{KL}}\left(\frac{\boldsymbol{v}_i}{\|\boldsymbol{v}_i\|_1} \Big\| \frac{\mathbf{J}(\pi_\theta)}{\|\mathbf{J}(\pi_\theta)\|_1}\right) \geq \psi, \quad (6)$$

where $\Psi(\pi_\theta, \boldsymbol{v}_i)$ defines a similarity metric between the reference vector $\boldsymbol{v}_i$ and the objective vector $\mathbf{J}(\pi_\theta) > \mathbf{0}$. When $\psi$ is large, $\mathbf{J}(\pi_\theta)$ is constrained to be close to $\boldsymbol{v}_i$, and the targeted region on the Pareto front is small. Hence, solving the constrained bi-objective optimization for the diverse set of reference vectors produces a diverse set of Pareto policies associated with different trade-offs between $J^{\hat{r}}$ and $J^u$. In the following, we develop an efficient two-stage algorithm for solving this optimization problem in Eq. (6).

To solve the bi-objective optimization with an inequality constraint in Eq. (6), we propose a two-stage gradient-based method in Alg. 2. It first finds a solution meeting the constraint by alternating between optimizing the two objectives (correction stage) and then treats the constraint as the third objective and applies an existing multi-objective optimization algorithm to find a Pareto policy in the region targeted by the constraint (ascending stage). In the first stage, the algorithm checks how the constraint is violated (line 3), e.g., whether $J^{\hat{r}}$ or $J^u$ is too small, and then accordingly chooses one to apply gradient ascent towards meeting the constraint. Once it finds a feasible solution fulfilling the constraint, it switches to the second stage, which reframes the problem as a tri-objective optimization by turning the constraint to be the third objective, i.e., $\max_\theta \mathbf{F}(\pi_\theta) \triangleq$

---

**Algorithm 2** A two-stage method for solving constrained bi-objective optimization

1: **input:** $\pi_{\theta_t}, \boldsymbol{v}_i, \psi$
2: **if** $\Psi(\pi_{\theta_t}, \boldsymbol{v}_i) < \psi$ **then**  $\quad \triangleright$ Correction stage
3: $\quad$ **if** $J^{\hat{r}}(\pi_{\theta_t})/J^u(\pi_{\theta_t}) < v_i^1/v_i^2$ **then**
4: $\quad\quad$ Compute $\nabla_\theta J^{\hat{r}}(\pi_{\theta_t})$;
5: $\quad\quad$ $\theta_{t+1} = \theta_t + \eta \nabla_\theta J^{\hat{r}}(\pi_{\theta_t})$;
6: $\quad$ **else**
7: $\quad\quad$ Compute $\nabla_\theta J^u(\pi_{\theta_t})$;
8: $\quad\quad$ $\theta_{t+1} = \theta_t + \eta \nabla_\theta J^u(\pi_{\theta_t})$;
9: **else**  $\quad\quad\quad\quad\quad\quad\quad \triangleright$ Ascending stage
10: $\quad$ Compute $\nabla_\theta \mathbf{F}(\pi_{\theta_t})$;
11: $\quad$ Find $\boldsymbol{\alpha}_t^*$ to Eq. (7);
12: $\quad$ $\theta_{t+1} = \theta_t + \eta \boldsymbol{\alpha}_t^* \nabla_\theta \mathbf{F}(\pi_{\theta_t})$;
13: $t \leftarrow t + 1$
14: **output:** $\pi_{\theta_t}$

---

$(J^{\hat{r}}(\pi_\theta), J^u(\pi_\theta), \Psi(\pi_\theta, \boldsymbol{v}_i))^\mathsf{T}$, and applies MGDA (Désidéri, 2012) to find a Pareto solution for this problem. Each step of MGDA aims at finding a convex combination $\boldsymbol{\alpha}_t \nabla_\theta \mathbf{F}(\pi_{\theta_t})$ of all objectives' gradients $\nabla_\theta \mathbf{F}(\pi_{\theta_t}) \triangleq (\nabla_\theta J^{\hat{r}}(\pi_{\theta_t}), \nabla_\theta J^u(\pi_{\theta_t}), \nabla_\theta \Psi(\pi_{\theta_t}, \boldsymbol{v}_i))^\mathsf{T}$ such that no objective is decreasing on the combined direction. This is equal to solving the following min-norm problem.

$$\min_{\boldsymbol{\alpha}_t} \|\boldsymbol{\alpha}_t \nabla_\theta \mathbf{F}(\pi_{\theta_t})\|_2, \text{s.t.} \|\boldsymbol{\alpha}\|_1 = 1, \boldsymbol{\alpha}_t \geq \mathbf{0}, \quad (7)$$

which can be efficiently solved by Frank-Wolfe's algorithm (Jaggi, 2013). Then MGDA takes one step along the direction to update the policy $\theta_t$, i.e., $\theta_{t+1} = \theta_t + \eta \boldsymbol{\alpha}_t \nabla_\theta \mathbf{F}(\pi_{\theta_t})$. Since MGDA always improves each objective, the constraint will not be violated, and the algorithm will finally find a Pareto policy in the targeted region. In practice, we use OpenAI's ES (Salimans et al., 2017) to estimate the gradient of each objective (see Appendix A.3) for its efficiency and stability. Alg. 2 provably converges to the Pareto front region targeted by the reference vector with approximate gradients.

**Assumption 1.** *Suppose $m$ objectives $\{f_i\}_{i=1}^m$ of a multi-objective function are differentiable and that their gradients are Lipschitz continuous with constant $L_i > 0$.*

**Assumption 2.** *For the ES gradient $\nabla f_{i,\nu}(x_t)$, we have $\mathbb{E}_{\varepsilon_t \sim \mathcal{N}(0,I)}[\nabla f_{i,\nu}(x_t)] = \nabla f_i(x_t)$. Suppose $\mathrm{Var}(\nabla f_{i,\nu}(x_t)) \leq \sigma^2$.*

**Lemma 1.** $\forall$ *mutually independent objectives $f_i \geq 0$, which satisfies Assumption 1 & 2, has $\mathbb{E}_{\varepsilon_t}[f_i(x_{t+1})] - f_i(x_t) \leq -(\eta - \frac{L_i \eta^2}{2})\|\bar{d}_t\|^2 + \frac{L_i \eta^2}{2}\sigma^2$ where $\bar{d}_t = \sum_{i=1}^m \alpha_{i,t} \nabla f_i(x_t)$.*

Lemma 1 implies that when $\eta < \frac{\|\bar{d}_t\|^2}{\|\bar{d}_t\|^2 + \sigma^2} \frac{2}{L_i}$, the ascending stage leads to a monotonically non-increasing sequence of the objectives.

**Theorem 1** (Non-convex convergence rate). *Let Assumption 1 & 2 hold,* $\Delta = f_i(x_0) - f_i(x^*)$, $\beta = \eta - \frac{L_i\eta^2}{2}$, *and* $\gamma = \frac{L_i\eta^2}{2}$. *For an arbitrary objective* $f_i$, *given any* $\epsilon > 0$, *after* $T = O(\frac{\Delta}{\beta\epsilon - \gamma\sigma^2})$ *iterations of ascending stage, we have* $\frac{1}{T}\sum_{t=0}^{T-1}\mathbb{E}_{x_t}[\|\bar{d}_t\|^2] \leq \epsilon$.

Complete proofs are provided in Appendix A.1. Theorem 1 provides the convergence rate of the ascending stage for non-convex objectives. For the correction stage that performs a single-objective ES gradient ascent, Nesterov & Spokoiny (2017) have proved its convergence rate to a stationary point of the non-convex objective.

### 4.2.2 LOCAL EXTENSION OF PARETO FRONT

Although solving Eq. 6 for a diverse set of reference vectors using the algorithm in Sec. 4.2.1 can produce a diverse set of Pareto policies, the computational cost linearly increases with the number of policies. In practice, a few Pareto policies cannot cover all possible trade-offs and thus may lead to sub-optimal choices of policy in deployment. In order to efficiently obtain more policies with different fine-grained levels of model return-uncertainty trade-off, we propose a "Pareto Extension" that starts from the diverse set of Pareto policies and locally searches for more policies near them on the Pareto front. This warm-start strategy avoids training more policies from scratch and practically saves a significant amount of computation. As illustrated in Fig. 2 and Alg. 1, we perturb each reference vector $\boldsymbol{v}_i$ for more reference vectors $\boldsymbol{v}_i^+$ and $\boldsymbol{v}_i^-$, i.e.,

$$\boldsymbol{v}_i^+ = \boldsymbol{v}_i + \boldsymbol{\epsilon}, \boldsymbol{v}_i^- = \boldsymbol{v}_i - \boldsymbol{\epsilon}, \boldsymbol{\epsilon} \triangleq (\epsilon, -\epsilon), 0 < \epsilon < \tau_a. \tag{8}$$

These new vectors create more constrained bi-objective optimization problems in the same form as Eq. (6). Instead of using a random initialization, we start from and fine-tune $\{\pi_i\}_{i=1}^n$ for a few iterations to achieve the Pareto policies of these new optimization problems. These policies and $\{\pi_i\}_{i=1}^n$ together constitute a pool of Pareto policies. When deployed to a realistic environment, we first evaluate these policies for a few steps and then select the one achieving the highest return to deploy. More details about our selection strategy are given in Appendix A.4. As long as we include sufficient diverse policies in the pool, we can find a promising policy that significantly outperforms the policies trained by other model-based offline RL methods.

## 5 EXPERIMENTS

This section aims to answer the following questions by evaluating P3 with other offline RL methods on the datasets from the D4RL Gym benchmark (Fu et al., 2020). **(1) Comparison to prior work**: Does P3 outperform other state-of-the-art offline RL methods? Moreover, when the dataset does not contain high-quality samples for some state-action pairs, can the policies generated by P3 generalize to unseen states in order to achieve high returns? **(2) Ablation study**: In the experiments, we apply several techniques used in previous work (see Appendix A.6 for more details). How do they affect performance? **(3) Effectiveness of P3**: Why is it challenging to find the optimal trade-off between the model return and its uncertainty? We empirically explain how P3 efficiently alleviate this problem by generating a rich pool of diverse and representative policies.

**To answer question (1)**, we compare P3's performance with the state-of-the-art offline RL algorithms, including BCQ (Fujimoto et al., 2019), BEAR (Kumar et al., 2019), CQL (Kumar et al., 2020), UWAC (Wu et al., 2021), TD3+BC (Fujimoto & Gu, 2021), MOPO (Yu et al., 2020), MOReL (Kidambi et al., 2020), and COMBO (Yu et al., 2021). For fairness of comparison, we re-run these algorithms using author-provided implementations[1] and train each algorithm for 1000 epochs. We also carefully tune the hyperparameters of baselines such as BCQ, CQL, TD3+BC, MOPO, and MOReL by grid search and choose the best ones for each benchmark dataset. Most of them achieve higher scores than those previous versions. For other baselines, such as UWAC and COMBO, we adopt the hyperparameters in their original papers, assuming they have chosen the best hyperparameters they can find. More details on the experimental setting can be found in Appendix A.6. Our experiment results are provided in Table 1. It is obvious that P3 achieves the highest average-score across all datasets, and significantly outperforms the baseline methods in 5 out of the 9 low/medium-quality datasets. Moreover, we find that the low/medium-quality datasets in the D4RL benchmark are "imbalanced". As illustrated in Fig. 8, there are a large number of bad samples, some mediocre

---

[1]As noted by https://github.com/aravindr93/mjrl/issues/35, we remark that the implementation provided by MOReL's author achieves lower results than their reported ones.

| | | BCQ | BEAR | CQL | UWAC* | TD3+BC | MOPO | MOPO* | MOReL | COMBO* | P3+FQE | P3 |
|---|---|---|---|---|---|---|---|---|---|---|---|---|
| Random | HalfCheetah | 2.2 ±0.1 | 2.3 ±0.1 | 21.7 ±0.6 | 14.5 ±3.3 | 10.6 ±1.7 | 35.9 ±2.9 | 35.4 ±2.5 | 30.3 ±5.9 | 38.8 | 37.4 ±5.1 | 40.6 ±3.7 |
| | Hopper | 8.1 ±0.5 | 3.9 ±2.3 | 8.1 ±1.4 | 22.4 ±12.1 | 8.6 ±0.4 | 16.7 ±12.2 | 11.7 ±0.4 | 44.8 ±4.8 | 17.9 | 33.8 ±0.4 | 35.4 ±0.8 |
| | Walker2d | 4.6 ±0.7 | 12.8 ±10.2 | 0.5 ±1.3 | 15.5 ±11.7 | 1.5 ±1.4 | 4.2 ±5.7 | 13.6 ±2.6 | 17.3 ±8.2 | 7.0 | 19.7 ±0.5 | 22.9 ±0.6 |
| Medium | HalfCheetah | 45.4 ±1.7 | 42.9 ±0.2 | 49.2 ±0.3 | 46.5 ±2.5 | 47.8 ±0.4 | 42.3 ±1.6 | 73.1 ±2.4 | 20.4 ±13.8 | 54.2 | 61.4 ±2.0 | 64.7 ±1.6 |
| | Hopper | 53.9 ±3.7 | 51.8 ±3.9 | 62.7 ±3.7 | 88.9 ±12.2 | 69.1 ±4.5 | 38.3 ±34.9 | 28.0 ±12.4 | 53.2 ±32.1 | 94.9 | 105.9 ±1.4 | 106.8 ±0.7 |
| | Walker2d | 74.5 ±3.7 | -0.2 ±0.1 | 57.5 ±8.3 | 57.5 ±7.8 | 81.3 ±3.0 | 41.2 ±30.8 | 17.8 ±19.3 | 10.3 ±8.9 | 75.5 | 71.1 ±3.5 | 81.3 ±2.0 |
| Medium-replay | HalfCheetah | 40.9 ±1.1 | 36.3 ±3.1 | 47.2 ±0.4 | 46.8 ±3.0 | 44.8 ±0.5 | 69.2 ±1.1 | 53.1 ±2.0 | 31.9 ±6.1 | 55.1 | 43.4 ±1.1 | 48.2 ±0.6 |
| | Hopper | 40.9 ±16.7 | 52.2 ±19.3 | 28.6 ±0.9 | 39.4 ±6.1 | 57.8 ±17.3 | 32.7 ±9.4 | 67.5 ±24.7 | 54.2 ±32.1 | 73.1 | 89.5 ±2.0 | 94.6 ±1.4 |
| | Walker2d | 42.5 ±13.7 | 6.9 ±7.8 | 45.3 ±2.7 | 27.0 ±6.3 | 81.9 ±2.7 | 73.7 ±9.4 | 39.0 ±9.6 | 13.7 ±8.1 | 56 | 60.1 ±9.5 | 64.0 ±8.2 |
| | Mean | 34.8 ±4.7 | 23.2 ±5.2 | 35.6 ±2.2 | 39.8 ±7.2 | 44.8 ±3.5 | 42.8 ±12.1 | 34.3 ±8.3 | 30.7 ±13.3 | 52.5 | 58.0 ±2.8 | 62.1 ±2.2 |
| Expert | HalfCheetah | 92.7 ±2.5 | 92.7 ±0.6 | 97.5 ±1.8 | 128.6 ±2.9 | 96.3 ±0.9 | 81.3 ±21.8 | – | 2.2 ±5.4 | – | 81.4 ±1.72 | 88.8 ±0.4 |
| | Hopper | 105.3 ±8.1 | 54.6 ±21.1 | 105.4 ±5.9 | 135.0 ±14.1 | 109.5 ±4.1 | 62.5 ±28.9 | – | 26.2 ±13.9 | – | 110.6 ±1.2 | 111.3 ±0.5 |
| | Walker2d | 109.1 ±0.4 | 106.8 ±6.8 | 108.9 ±0.4 | 121.1 ±22.4 | 110.3 ±0.4 | 62.4 ±3.2 | – | -0.3 ±0.3 | – | 102.0 ±3.4 | 106.7 ±0.2 |
| Medium-expert | HalfCheetah | 93.9 ±1.2 | 46.1 ±4.7 | 70.6 ±13.6 | 127.4 ±3.7 | 88.9 ±5.3 | 70.3 ±21.9 | 63.3 ±38.0 | 35.9 ±19.2 | 90 | 57.1 ±16.0 | 69.9 ±10.5 |
| | Hopper | 108.6 ±5.9 | 50.6 ±25.3 | 111.0 ±1.2 | 134.7 ±21.2 | 102.0 ±10.1 | 60.6 ±32.5 | 23.7 ±6.0 | 52.1 ±27.7 | 111.1 | 109.4 ±1.3 | 110.8 ±0.5 |
| | Walker2d | 109.7 ±0.6 | 22.1 ±44.5 | 109.7 ±0.3 | 99.7 ±12.2 | 110.5 ±0.3 | 77.4 ±27.9 | 44.6 ±12.9 | 3.9 ±2.8 | 96.1 | 90.3 ±4.2 | 98.9 ±3.4 |
| | Mean | 103.2 ±3.1 | 62.2 ±17.2 | 100.5 ±3.9 | 124.4 ±12.8 | 102.9 ±3.5 | 69.1 ±22.7 | 43.9 ±19.0 | 20.0 ±7.7 | 99.1 | 91.8 ±4.6 | 97.7 ±2.6 |
| Total | Mean | 62.2 ±4.1 | 38.8 ±10.0 | 61.6 ±2.9 | 73.6 ±9.4 | 68.0 ±3.5 | 53.3 ±16.3 | 36.8 ±11.0 | 26.4 ±11.1 | 64.2 | 71.5 ±3.5 | 76.3 ±2.4 |

Table 1: **Results on D4RL Gym experiments.** Normalized score (mean±std) over the final 10 evaluations and 5 seeds. ∗ marks previously reported results. Dataset quality gradually improves from Random to Medium-expert.

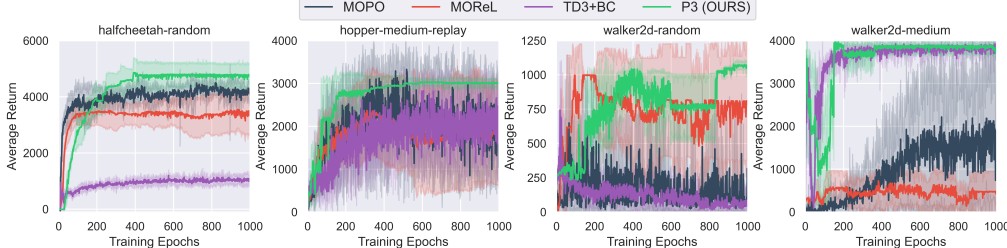

Figure 3: **Learning curves on low-quality datasets.** Returns are averaged over 10 evaluations and 5 seeds. The shaded area depicts the standard deviation over 5 seeds. P3 outperforms two recent model-based offline RL methods (i.e., MOPO and MOReL) and the SoTA model-free method (i.e, TD3+BC). A full results of all datasets are in Fig. 7 of Appendix.

samples, and a few good samples, causing problems with learning accurate behavior policies or generalizable environment models (Buckman et al., 2021; Zhang et al., 2021). Therefore, many offline RL algorithms, especially model-free algorithms that heavily rely on the accurate recovery of behavior policy (Fujimoto et al., 2019; Kumar et al., 2020; Fujimoto & Gu, 2021), perform poorly on these datasets. According to the results in Table 1, P3 achieves SoTA performance and outperforms other baselines by a large margin on random, medium, and medium-replay datasets, indicating the advantages of P3 on learning from low-quality experiences. We use online policy evaluation to select the best policy during the test phase of P3, which can be computationally intensive or cause overheads when deployed to realistic environments. To overcome this drawback, we replace the online evaluation with Fitted Q Evaluation (FQE) (Le et al., 2019), an offline policy evaluation method, which (approximately) evaluates policies using the offline data only. The implementation details and experimental results are reported in Appendix A.4 and Table 1, respectively. We surprisingly find that "P3+FQE (offline policy evaluation)" only slightly degrades from "P3+online policy evaluation" on the performance, but still outperforms all baselines on the low/medium datasets, suggesting that FQE enables more efficient inference for P3 so it has exactly the same inference cost as other baselines.

**To answer question (2)**, we conduct a thorough ablation study toward five variants of P3, each removing/changing one component used in P3. Table 2 reports their results on the D4RL Gym benchmark with 9 representative environment-dataset combinations. In Fig. 9 in Appendix, we visualize the Pareto policies obtained by P3 to highlight the effectiveness and superiority of our method. Among the five variants of P3, "scalarization" replaces our posposed Alg. 2 with the scalarization method (Boyd & Vandenberghe, 2004, Chapter 4.7); "no StateNorm" removes the state normalization (Mania et al., 2018; Fujimoto & Gu, 2021); "no RankShaping" removes the rank-based scale shaping (Wierstra et al., 2014); "no ParetoExtension" removes the Pareto extension proposed in Sec. 4.2.2; "no BehaviorCloning" removes the behavior cloning initialization (Kumar et al., 2020; Kidambi et al., 2020). More details on these variants are provided in Appendix A.9. According to the results in Table 2 and Fig. 9, we give the following conclusions: (1) except "scalarization" and "no ParetoExtension", other variants perform comparably to our P3 while outperforming the previous results achieved by model-based (MOPO) and model-free (TD3+BC) RL algorithms on

| Data Quality | Random | | | Medium-replay | | | Medium-expert | | |
|---|---|---|---|---|---|---|---|---|---|
| Environment | HalfCheetah | Hopper | Walker2d | HalfCheetah | Hopper | Walker2d | HalfCheetah | Hopper | Walker2d |
| P3: scalarization | 15.5 ±0.8 | 32.3 ±1.5 | 15.2 ±5.0 | 40.1 ±1.4 | 88.5 ±8.3 | 49.9 ±15.0 | 52.4 ±7.3 | 77.3 ±22.9 | 84.7 ±8.5 |
| P3: no StateNorm | 35.3 ±2.5 | 34.9 ±0.2 | 21.8 ±0.3 | 41.7 ±0.4 | 82.3 ±12.9 | 61.6 ±9.4 | 47.1 ±0.3 | 99.9 ±6.0 | 90.3 ±2.2 |
| P3: no RankShaping | 37.6 ±4.4 | 33.6 ±0.3 | 27.3 ±6.2 | 44.3 ±0.7 | 95.6 ±1.7 | 64.7 ±3.9 | 66.3 ±1.9 | 108.3 ±1.2 | 97.0 ±2.6 |
| P3: no ParetoExtension | 31.2 ±2.4 | 5.2 ±0.4 | 0.1 ±0.2 | 43.4 ±1.6 | 91.3 ±4.9 | 2.0 ±0.6 | 4.7 ±3.2 | 88.2 ±16.4 | 0.3 ±0.1 |
| P3: no BehaviorCloning | 38.2 ±1.4 | 35.5 ±0.5 | 24.1 ±1.1 | 45.4 ±1.8 | 97.1 ±2.1 | 26.1 ±4.9 | 52.2 ±3.5 | 89.8 ±16.6 | 69.1 ±9.1 |
| P3: our version | 40.6 ±3.7 | 35.4 ±0.8 | 22.9 ±0.6 | 48.2 ±0.6 | 94.6 ±1.4 | 64.0 ±8.2 | 69.9 ±10.5 | 110.8 ±0.5 | 98.9 ±3.4 |
| MOPO | 35.9 ±2.9 | 16.7 ±12.2 | 4.2 ±5.7 | 69.2 ±1.1 | 32.7 ±9.4 | 73.7 ±9.4 | 70.3 ±21.9 | 60.6 ±32.5 | 77.4 ±27.9 |
| TD3+BC | 10.6 ±1.7 | 8.6 ±0.4 | 1.5 ±1.4 | 44.8 ±0.5 | 57.8 ±17.3 | 81.9 ±2.7 | 88.9 ±5.3 | 102.0 ±10.1 | 110.5 ±0.3 |

Table 2: **Ablation study.** Normalized score (mean±std) of P3 variants over the final 10 evaluations and 5 seeds when applied to three representative D4RL datasets, i.e., random, medium-replay, and medium-expert, corresponding to low, medium and high-quality data, respectively.

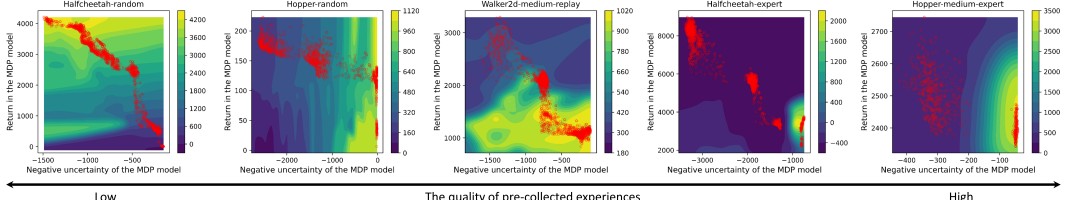

Figure 4: Model-based offline RL's performance in the deployed environment (heatmap) under different trade-offs between the model return (y-axis) and uncertainty (x-axis). Each red circle is a Pareto policy from the pool generated by P3. Zoom in for more details. More results are shown in Fig. 10 of Appendix.

the low/medium-quality datasets, reflecting that these widely-used techniques can improve P3's performance but are not crucial to our appealing results. (2) "scalarization" shows noticeable degradation in performance and cannot obtain a dense set of diverse policies, as shown in Fig. 9. The results can be explained as follows: the scalarization method only finds a few separated policies, and it is difficult to find one with advantageous trade-off among them. In addition, we remark that the computational cost of multiple training with different weight assignments is similar to the cost of running P3. (3) "no ParetoExtension" degrades P3's performance on all 9 environment-dataset combinations, corroborating that a dense set of policies on the Pareto front is essential to our results.

**To answer question (3)**, in Fig. 4, we study how the P3 policies with different uncertainty-return trade-off perform in the deployed environment. For low/medium-quality datasets (the left three plots in Fig. 4), the optimal policies with high realistic returns (bright areas in the heatmap) spread across almost the whole Pareto front. Therefore, to find the best policy, it is essential to explore the whole Pareto front and select one from a diverse set of Pareto optimal/stationary policies as P3 does. This explains why P3 performs the best on all the low/medium datasets. On the contrary, for high-quality datasets (the right two plots in Fig. 4), the optimal policies with high realistic returns gather within a small region of the Pareto front and associate to one trade-off level. Therefore, by carefully tuning the trade-off weight, previous methods can still find the optimal policy without visiting the whole Pareto front. Hence, we observe less advantages of P3 on the high-quality datasets. The reason behind is that the MDP models are very confident on high-return (realistic) state-action pairs if most samples in the training data are with high-return (high-quality), while they can be uncertain about many high-return pairs if the training data only cover a few high-return samples (low/medium quality). It is worth noting that collecting high-quality datasets is usually expensive or infeasible in practice and many applications lack sufficient high-quality data. In these imperfect but practical scenarios, P3 performs significantly better and more stably than existing model-based offline RL methods that only learns one single policy.

## 6 CONCLUSION

In this paper, we find that model-based offline RL's performance significantly relies on the trade-off between model return and its uncertainty, while determining the optimal trade-off is challenging without access to the realistic environment. To address the problem, we study a bi-objective formulation for model-based offline RL and develop an efficient method that produces a pool of diverse policies on the Pareto front performing different levels of trade-offs, which provides flexibility to select the best policy in the inference stage. We extensively validate the efficacy of our method on the D4RL benchmark, where ours largely outperforms several recent baselines and exhibits promising results on low-quality datasets.

## ACKNOWLEDGMENTS

Yijun Yang and Yuhui Shi are supported in part by the Shenzhen Fundamental Research Program under Grant No. JCYJ20200109141235597, the National Science Foundation of China under Grant No. 61761136008, the Shenzhen Peacock Plan under Grant No. KQTD2016112514355531, the Program for Guangdong Introducing Innovative and Entrepreneurial Teams under Grant No. 2017ZT07X386.

## REPRODUCIBILITY STATEMENTS

Code is available at https://github.com/OverEuro/P3. We provide a full description of all our experiments in Section 5 and Appendix A.6.

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

# A APPENDIX

## A.1 CONVERGENCE ANALYSIS OF ALG. 2

We provide a proof for Lemma 1.

*Proof.* For $\boldsymbol{\alpha}_t$ obtained by solving $\min_{\boldsymbol{\alpha}_t} \| \sum_{i=1}^m \alpha_{i,t} \nabla f_{i,\nu}(x_t) \|_2$, s.t. $\sum \alpha_t^i = 1, \alpha_t^i \geq 0$, let

$$x_{t+1} = x_t - \eta d_t$$
$$= x_t - \eta \sum_{i=1}^m \alpha_{i,t} \nabla f_{i,\nu}(x_t) \tag{9}$$

According to assumption 1:

$$f_i(x_{t+1}) \leq f_i(x_t) + \nabla f_i(x_t)^\mathsf{T}(x_{t+1} - x_t) + \frac{L_i}{2}\|x_{t+1} - x_t\|^2$$
$$= f_i(x_t) - \eta \nabla f_i(x_t)^\mathsf{T} d_t + \frac{L_i}{2}\|\eta d_t\|^2 \tag{10}$$

Take expectation on both sides:

$$\mathbb{E}_{\varepsilon_t}[f_i(x_{t+1})] \leq f_i(x_t) - \eta \nabla f_i(x_t)^\mathsf{T} \bar{d}_t + \frac{L_i \eta^2}{2}\mathbb{E}_{\varepsilon_t}[\|d_t\|^2] \qquad (\bar{d}_t = \sum_{i=1}^m \alpha_{i,t} \nabla f_i(x_t))$$

$$= f_i(x_t) - \eta \nabla f_i(x_t)^\mathsf{T} \bar{d}_t + \frac{L_i \eta^2}{2}(\|\mathbb{E}_{\varepsilon_t}[d_t]\|^2 + \mathrm{Var}(d_t))$$

$$\leq f_i(x_t) - \eta\|\bar{d}_t\|^2 + \frac{L_i \eta^2}{2}(\|\bar{d}_t\|^2 + \mathrm{Var}(d_t)) \qquad ((\text{Désidéri, 2012, Lemma 1}))$$

$$= f_i(x_t) - \eta\|\bar{d}_t\|^2 + \frac{L_i \eta^2}{2}(\|\bar{d}_t\|^2 + \sum_{i=1}^m \alpha_{i,t}^2 \mathrm{Var}(\nabla f_{i,\nu}(x_t)))$$

$$\leq f_i(x_t) - \eta\|\bar{d}_t\|^2 + \frac{L_i \eta^2}{2}(\|\bar{d}_t\|^2 + \sum_{i=1}^m \alpha_{i,t} \mathrm{Var}(\nabla f_{i,\nu}(x_t)))$$

$$\leq f_i(x_t) - (\eta - \frac{L_i \eta^2}{2})\|\bar{d}_t\|^2 + \frac{L_i \eta^2}{2}\sigma^2 \tag{11}$$

Hence, we have $\mathbb{E}_{\varepsilon_t}[f_i(x_{t+1})] - f_i(x_t) \leq -(\eta - \frac{L_i \eta^2}{2})\|\bar{d}_t\|^2 + \frac{L_i \eta^2}{2}\sigma^2$.

Let the right term:

$$-(\eta_t - \frac{L_i \eta_t^2}{2})\|\bar{d}_t\|^2 + \frac{L_i \eta_t^2}{2}\sigma^2 < 0 \tag{12}$$

We have:

$$\eta_t < \frac{\|\bar{d}_t\|^2}{\|\bar{d}_t\|^2 + \sigma^2} \frac{2}{L_i} \tag{13}$$

When $\eta_t < \frac{\|\bar{d}_t\|^2}{\|\bar{d}_t\|^2 + \sigma^2} \frac{2}{L_i}$, $-(\eta_t - \frac{L_i \eta_t^2}{2})\|\bar{d}_t\|^2 + \frac{L_i \eta_t^2}{2}\sigma^2 < 0$, then $\mathbb{E}_{\varepsilon_t}[f_i(x_{t+1})] - f_i(x_t) \leq 0$. Hence, the ascending stage leads to a monotonically non-increase sequence, convergence proved. $\square$

We provide a proof for Theorem 1.

*Proof.* According to Lemma 1, we have:

$$f_i(x_t) - \mathbb{E}_{\varepsilon_t}[f_i(x_{t+1})] \geq \beta\|\bar{d}_t\|^2 - \gamma\sigma^2 \tag{14}$$

Take expectation with respect to the solution $x_t$ and do telescoping:

$$\mathbb{E}_{x_0}[f_i(x_0)] - \mathbb{E}_{x_0,\varepsilon_0}[f_i(x_1)] \geq \beta\mathbb{E}_{x_0}[\|\bar{d}_0\|^2] - \gamma\sigma^2$$
$$\mathbb{E}_{x_1}[f_i(x_1)] - \mathbb{E}_{x_1,\varepsilon_1}[f_i(x_2)] \geq \beta\mathbb{E}_{x_1}[\|\bar{d}_1\|^2] - \gamma\sigma^2$$
$$\cdots$$
$$\mathbb{E}_{x_{T-1}}[f_i(x_{T-1})] - \mathbb{E}_{x_{T-1},\varepsilon_{T-1}}[f_i(x_T)] \geq \beta\mathbb{E}_{x_{T-1}}[\|\bar{d}_{T-1}\|^2] - \gamma\sigma^2 \tag{15}$$

Note that $\mathbb{E}_{x_T, \varepsilon_T}[f_i(x_{T+1})] = \mathbb{E}_{x_{T+1}}[f_i(x_{T+1})]$, hence

$$\Delta = f_i(x_0) - f_i(x^*)$$
$$\geq f_i(x_0) - \mathbb{E}_{x_T}[f_i(x_T)] \qquad\qquad (x^* \text{ dominates } x_T)$$

$$\geq T(\beta \frac{1}{T} \sum_{t=0}^{T-1} \mathbb{E}_{x_t}[\|\bar{d}_t\|^2] - \gamma\sigma^2) \tag{16}$$

According to Eq. 16, we can obtain

$$\frac{1}{T} \sum_{t=0}^{T-1} \mathbb{E}_{x_t}[\|\bar{d}_t\|^2] \leq (\frac{\Delta}{T} + \gamma\sigma^2)\frac{1}{\beta} \tag{17}$$

Let

$$(\frac{\Delta}{T} + \gamma\sigma^2)\frac{1}{\beta} \leq \epsilon \tag{18}$$

Rearranging terms, we have

$$T \geq \frac{\Delta}{\beta\epsilon - \gamma\sigma^2} \tag{19}$$

By the condition that $\beta\epsilon - \gamma\sigma^2 > 0$, we have the following:

$$\epsilon(\eta - \frac{L_i\eta^2}{2}) - \frac{L_i\eta^2}{2}\sigma^2 > 0$$
$$\epsilon - \frac{L_i\eta}{2}\epsilon - \frac{L_i\eta}{2}\sigma^2 > 0$$
$$\frac{\epsilon}{\epsilon + \sigma^2}\frac{2}{L_i} > \eta \tag{20}$$

Let $\eta < \frac{\epsilon}{\epsilon+\sigma^2}\frac{2}{L_i}$, given $\epsilon > 0$, after $T = O(\frac{\Delta}{\beta\epsilon-\gamma\sigma^2})$ iterations, $\frac{1}{T}\sum_{t=0}^{T-1} \mathbb{E}_{x_t}[\|\bar{d}_t\|^2] \leq \epsilon$. $\qquad\square$

## A.2 DETAILED RELATED WORK

**Model-based Offline RL.** Model-based RL methods are promising candidates for sequential decision-making problems due to their high sample-efficiency and compact modeling of a dynamic environment. Time-dependent linear models and Gaussian processes provide excellent performance in the low-data and low-dimensional scenario. High-capacity models, e.g., deep neural networks, are more successful because they benefit from powerful supervised learning techniques that allow the leverage of large-scale datasets, even in domains with high-dimensional image observations. Although the convenience of working with large-scale datasets, these methods still suffer from the effects of distribution shift and error accumulation of the model predictions (Levine et al., 2020), especially in the offline setting. Existing works explore several classical methods to solve the problem. Dyna-style algorithms (Sutton, 1991) utilize a technique named *branched rollout* to collect short-horizon model rollouts, which alleviates accumulated extrapolation errors. Another method introduces ensemble learning (Chua et al., 2018), i.e., the leverage of multiple dynamics models, against model errors. Other works also incorporate novel concepts into model-based RL, e.g., an energy-based model regularization (Boney et al., 2019), a game-theoretic framework for model-based RL (Rajeswaran et al., 2020), meta-learning (Clavera et al., 2018), policy regularization (Berkenkamp et al., 2017), and generative temporal difference learning (Janner et al., 2020). In the offline setting, since the learned model cannot be corrected with additional data collection and training, it is crucial to prevent the policy from visiting "out-of-distribution" states. Recent two works propose uncertainty-regularized policy optimization algorithms for this purpose (Yu et al., 2020; Kidambi et al., 2020). One adds a soft uncertainty penalty associated with a user-chosen weight to the reward issued by the model, and the other constructs a pessimistic MDP model associated with a hard threshold for terminating the interaction between the policy and the model when the model prediction is inaccurate. (Matsushima et al., 2021) presents a deployment-efficient RL framework to avoid the offline distribution shift issue. (Rafailov et al., 2021) extends an existing model-based offline RL method to image-based tasks. More recently, conservative offline model-based policy optimization (COMBO) (Yu et al., 2021) regularizes the value function on out-of-distribution states yielded via interacting with the learned model, which leads to a conservative estimate of value function for these state-action pairs, without requiring explicit uncertainty penalty. Moreover, a recent method named *MuZero Unplugged* focuses on using the model-based planning directly for policy improvement (Schrittwieser et al., 2021).

**Multi-objective Optimization.** Multi-objective optimization (MOO), also known as Pareto optimization, aims to optimize more than one objective function. For a nontrivial MOO problem, no single solution exists that optimizes all objectives simultaneously. In this case, these objective functions are said to be conflicting, and we can find a set of Pareto optimal solutions given different trade-offs among objectives. Existing methods in MOO can be divided into two categories: gradient-based methods and heuristic methods. Multiple gradient descent algorithm (MGDA) (Désidéri, 2012) is an extension of the classical gradient descent algorithm to multiple objectives. It uses an adaptive weighting aggregation of sub-objective gradients to compute the descent direction for all objectives. Because of the ability of fast convergence with theoretical guarantee, MGDA-based methods have been widely used to solve multi-task learning (Sener & Koltun, 2018; Lin et al., 2019) and multi-objective recommendation (Milojkovic et al., 2020). However, MGDA-based methods generate only one solution or a finite set of sparse solutions with different trade-offs. In contrast, heuristic methods usually find a Pareto set by various evolutionary algorithms such as the genetic algorithm (NSGA-II) (Deb et al., 2002) and the decomposition-base evolution algorithm (MOEA/D) (Zhang & Li, 2007). They can find a dense set of Pareto near-optimal solutions by only one time running. But in practice, these black-box optimization methods usually take unacceptable run time when the parameter space is extremely large, e.g., the number of parameters for a deep neural network, or when the function evaluation is computationally expensive, e.g., population-based training for RL agents. In this paper, our approach borrows the ideas from the heuristic methods to achieve a dense set of policies. Moreover, we replace inefficient evolutionary operators with a gradient-based method, reducing the training cost significantly.

**Model-free Offline RL.** Model-free RL directly learn a policy without requiring the learning of an environment model. A direct extension of model-free RL methods to offline setting (Fujimoto et al., 2019) usually perform poorly due to the data distribution shift. To address this issue, prior model-free offline RL methods regularize the learned policy to be "close" to the behavior policy either implicitly by conservative Q learning (Riedmiller, 2005; Fujimoto et al., 2019; Kumar et al., 2020; Zhang et al., 2021; Fujimoto & Gu, 2021), or explicitly by direct constraints in state or action spaces (Kumar et al., 2019; Liu et al., 2020; Zhou et al., 2020). Compared with model-based offline RL methods, these model-free methods behave more conservatively and lack broader generalization when the distribution of pre-collected experiences is narrow. Instead of formulating as a single-objective optimization with regularization, a concurrent work (DiME) by Abdolmaleki et al. (2021) takes a multi-objective perspective for model-free offline RL. It adopts a modified scalarization method to achieve multiple policies on the Pareto front. However, scalarization cannot find any solution on the non-concave parts of Pareto front and thus may lead to sub-optimal performance.

## A.3 Evolution Strategy

For black-box or noisy objective functions such as RL's policy optimization, computing accurate the gradient is usually challenging. Hence, as a derivative-free optimization method, evolution strategy (ES) has seen a recent revival in the RL community (Salimans et al., 2017; Mania et al., 2018). Instead of solving the complicated objective function $\mathbf{F}(\theta)$ directly, ES solves its Gaussian smoothing version: $\mathbf{F}_\nu(\theta) = \mathbb{E}_{\varepsilon \sim \mathcal{N}(0,I)}[\mathbf{F}(\theta + \nu\varepsilon)]$, where $\nu > 0$ denotes a smoothing parameter. When $\nu$ is small, the smoothed version $\mathbf{F}_\nu(\theta)$ is close to the original objective (Nesterov & Spokoiny, 2017), and its gradient with respect to policy parameters $\theta$ is defined by $\nabla_\theta \mathbf{F}_\nu(\theta) = (2\pi)^{-d/2} \int_{\mathbb{R}^d} \mathbf{F}(\theta + \nu\varepsilon)e^{-\frac{1}{2}\|\varepsilon\|_2^2}\varepsilon\mathrm{d}\varepsilon$. Although the gradient is intractable, it can be estimated by a standard Monte Carlo method: $\nabla_\theta \mathbf{F}_\nu(\theta) = \frac{1}{k\nu}\sum_{i=1}^k \mathbf{F}(\theta + \nu\varepsilon_i)\varepsilon_i$. The Monte Carlo estimation has no bias but high variance. Many following algorithms were proposed to reduce the variance while keeping the bias unchanged. In this paper, we adopt an antithetic estimator (Mania et al., 2018), as shown in Eq. (21), which uses the symmetric difference between a perturbation $\varepsilon_i \sim \mathcal{N}(\mathbf{0}, I)$ and its antithetic counterpart $-\varepsilon_i$.

$$\nabla_\theta \mathbf{F}_\nu(\theta) = \frac{1}{2k\nu}\sum_{i=1}^k [\mathbf{F}(\theta + \nu\varepsilon_i) - \mathbf{F}(x - \nu\varepsilon_i)]\varepsilon_i, \tag{21}$$

$$\mathbf{F}(\theta) = \left(J^{\hat{r}}(\theta), J^u(\theta), \Psi(\theta, \boldsymbol{v}_i)\right)^\top. \tag{22}$$

where $J^{\hat{r}}(\theta)$, $J^u(\theta)$ and $\Psi(\theta, \boldsymbol{v}_i)$ are results achieved by the policy $\pi_\theta$ on one trajectory generated from the environment model. Despite the simplicity, ES achieves competitive performance compared to policy gradient methods (Choromanski et al., 2020).

## A.4   SELECTION OF PARETO POLICIES BY FITTED Q EVALUATION (FQE)

Given a pool of Pareto policies, how do we determine the best one for a realistic environment? A straightforward method is online policy selection that evaluates the performance of every Pareto policy in the real environment, and then selects the one with the highest return. Most existing offline RL methods (Kidambi et al., 2020; Kumar et al., 2020; Matsushima et al., 2021; Zhang et al., 2021) adopt this method, which however requires many steps of online interactions with the realistic environment, especially when the number of policies to be evaluated is large. We also follow these works and use online policy evaluation to achieve the results in Table 1.

Probably a more efficient alternative is offline policy selection (Voloshin et al., 2019), which is proposed very recently, and aims at choosing the best policy from a set of policies, given only offline data. However, offline policy selection/evaluation is an open problem (Levine et al., 2020), and a number of recent works in RL (Paine et al., 2020; Zhang & Jiang, 2021; Fu et al., 2021) are still exploring new strategies for it while combining our method with offline policy evaluation methods can potentially improve the efficiency of P3. To evaluate this strategy, we use FQE (Le et al., 2019) to estimate the performance of each policy. Although there exist various offline policy evaluation methods, we choose FQE for its simplicity and stability in practice. By contrast, other methods have to solve complex estimation problems, such as learning a transition model from visual inputs or estimating the importance weights in a continuous action space (Fu et al., 2021). In our experiments, both P3 and FQE have access to the same offline data.

On an offline dataset $D$, FQE algorithm trains a critic $Q_{\phi_i}$ for each Pareto policy $\pi_i$ generated by P3, and then uses the critic to estimate the expected value of $\pi_i$ w.r.t. the initial state $s_0$, i.e., $\hat{V}_{\pi_i} = \mathbb{E}_{s_0 \sim D}[Q_{\phi_i}(s_0, \pi_i(s_0))]$, which reasonably quantifies the performance of $\pi_i$ when deployed to a realistic environment. The pseudo-code for FQE algorithm can be found in Alg. 3. FQE can efficiently find a near-optimal Pareto policy after only hundreds of epochs, which is demonstrated by the results in Appendix A.7.

---

**Algorithm 3** Fitted Q evaluation (FQE) for Pareto policy selection

1: **input:** Pareto policy pool $\mathcal{P}$, dataset $D$, $\gamma = 0.99$, $\beta = 0.995$
2: **for** $\pi_i \in \mathcal{P}$ (in parallel) **do**
3:     Initialize a critic $Q_{\phi_i}$ and the corresponding target critic $Q'_{\phi'_i} \leftarrow Q_{\phi_i}$;
4:     **for** $k = 1, 2, \ldots, K_{epoch}$ **do**
5:         Sample a batch $\{s_j, a_j, r_j, s'_j\}_{j=1}^{batchsize}$ from $D$;
6:         Update $Q_{\phi_i}$ by minimizing $\text{MSELoss}(Q_{\phi_i}(s, a), r + \gamma Q'_{\phi'_i}(s', \pi_i(s')))$;
7:         $\phi'_i = \beta \phi'_i + (1 - \beta)\phi_i$;                    ▷ Update target critic $Q'_{\phi'_i}$
8:     $\hat{V}_{\pi_i} = \mathbb{E}_{s_0 \sim D}[Q_{\phi_i}(s_0, \pi_i(s_0))]$;
9: **output:** $i_{best} = \arg\max_i \hat{V}_{\pi_i}$

---

## A.5   ENVIRONMENT MODEL

We train an ensemble of $N$ models and pick the best $K$ models based on their prediction error on a hold-out set. In the training phase, each model is optimized independently using the maximum likelihood estimation with mini-batch stochastic gradient descent. In the inference phase, we randomly select one of $K$ models and draw a state-reward concatenation from the resulting distribution, allowing for different transitions along a single episode to be sampled from different dynamics models. The method makes our model more uncertain and noisier than the realistic environment, resulting in a more challenging MDP problem. Prior work (Ha & Schmidhuber, 2018; Janner et al., 2019; Yu et al., 2020) demonstrated that the method effectively alleviates the model exploitation issue (Levine et al., 2020), especially in the offline setting. As shown in Table 3, we list the hyperparameters of environment model for D4RL Gym experiments.

## A.6   D4RL GYM EXPERIMENTS

**D4RL Gym Datasets.** D4RL is a widely-used benchmark for evaluating offline RL algorithms. It provides a variety of environments, tasks, and corresponding datasets containing samples of multiple trajectories generated via behavior policies, hand-designed controllers, or human demonstrators (Fu et al., 2020). In the continuous control domain based on the MuJoCo simulator (Todorov et al., 2012),

we apply a subset of datasets, including three environments (halfcheetah, hopper, and walker2d) and five dataset types (random, medium, medium-replay, expert, and medium-expert), to yield a total of 15 benchmark problems, in which **random** contains 1M samples from a random policy, **medium** contains 1M samples from a policy trained to approximately 1/3 of the performance of the expert, **expert** contains 1M samples from a policy trained to the performance of the expert, **medium-replay** contains the whole replay buffer of a policy trained up to the performance of the medium agent, and **medium-expert** contains a 50-50 split of medium and expert dataset (2M samples).

**Practical Modifications.** We list the techniques adopted by P3 for D4RL Gym experiments.

(1) P3 applies a simple behavior cloning approach (Kidambi et al., 2020; Matsushima et al., 2021; Kumar et al., 2020) to estimate the initial policy $\pi_i$. The technique effectively alleviates the data distribution shift caused by the difference between the policy-in-training and the behavior policies used to collect the data.

(2) We notice that the states of the high-dimensional complex tasks take the values in a broad range, causing the policies only pay attention to particular features of these states. Therefore, we normalize the received state before feeding it to the policy network: $a_h = \pi(\sigma_s^{-1}(\hat{s}_h - \mu_s))$ where $\mu_s$ and $\sigma_s$ are the mean and standard deviation of states, respectively. In the offline setting, we can easily obtain $\mu_s$ and $\sigma_s$ by computing the mean and standard deviation of all the states in the dataset, similar to the feature normalization in supervised learning. Previous work (Mania et al., 2018; Fujimoto & Gu, 2021) demonstrated that the technique makes the policies more robust to multiple-scale state inputs.

(3) P3 applies the rank-based scale shaping (Wierstra et al., 2014) to compute the ES gradients, which transforms the objective function value via a rank-based score function, suppressing the influence of outlier perturbations.

**Hyperparameters.** We outline the hyperparameters used in the experiments.

Table 3: Hyperparameters of environment model for D4RL Gym experiments.

| Hyperparameter | Value |
|---|---|
| Number of models/elites | 7/5 |
| Structure of hidden layers | $\text{MLP}(200, 200) \times 4$ |
| Nonlinearity function | Swish |
| Batch size | 256 |
| Optimizer | Adam |
| Learning rate | $10^{-4}$ |
| Weight decay | $10^{-5}$ |
| Holdout ratio | 0.1 |

Table 4: Hyperparameters of P3 for D4RL Gym experiments.

| Hyperparameter | Value |
|---|---|
| Policy network | $\text{MLP}(32, 32)$ |
| Nonlinearity function | Tanh |
| Step size $\eta$ | HalfCheetah: $2 \times 10^{-2}$ |
| | Hopper: $2 \times 10^{-2}$ |
| | Walker2d: $1.5 \times 10^{-2}$ |
| Number of perturbations $k$ (ES) | HalfCheetah: 30 |
| | Hopper: 30 |
| | Walker2d: 40 |
| Std. of perturbations $\nu$ (ES) | HalfCheetah: $3 \times 10^{-2}$ |
| | Hopper: $3 \times 10^{-2}$ |
| | Walker2d: $2.5 \times 10^{-2}$ |
| Horizon length $H$ | 1000 |
| Number of reference vectors $n$ | 5 |
| Number of updates $T = n(T_g + 2T_l)$ | $1000 = 5 \times (150 + 2 \times 25)$ |
| Range $(\tau_a, \tau_b)$ of reference vectors | $(0.1, 0.9)$ |
| Temperature $\kappa$ | 1.5 |
| Constraint threshold $\psi$ | $-10^{-3}$ |
| Local perturbation $\epsilon$ | $5 \times 10^{-2}$ |

### A.7 RANK CORRELATION BETWEEN OFFLINE AND ONLINE POLICY EVALUATION IN P3 ON D4RL GYM EXPERIMENTS

In Section A.4, in order to remove the extra computation required by online policy evaluation during the test, we instead use Fitted Q Evaluation (Alg. 3) to provide metrics for policy selection. Here we take a closer look at the correlation between the policy ranking by the online evaluation and the policy ranking by FQE.

First, we rank all the policies in the P3 pool by the two metrics and show their generated rank orders (in an ascending order) for every policy in Fig. 5. On three datasets, the two rank orders show a strong correlation, which indicates that the FQE based offline policy evaluation provides an accurate approximation to the realistic return of P3 policies, and thus can be used to improve the test efficiency of P3. In addition, we report the Spearman's rank correlation ("Spearman's $\rho$") and Kendall rank correlation ("Kendall's $\tau$") to quantitatively measure the correlation between the two rank orders. We report the results in Fig. 6, which demonstrates a strong correlation coefficient close to $1.0$ for all datasets. Therefore, replacing the online evaluation with FQE is an efficient solution preserving the original rank orders of P3 policies.

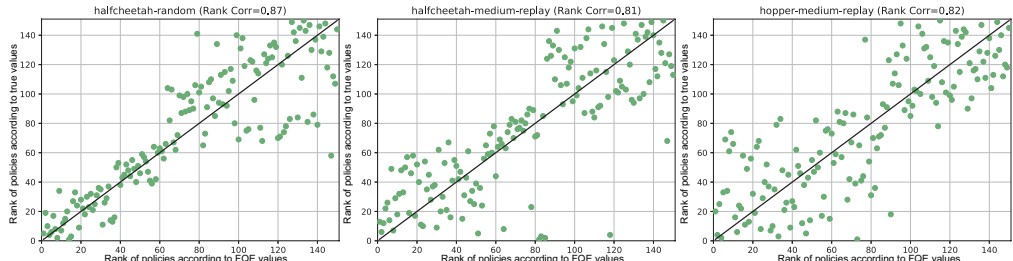

Figure 5: Scatter plots of the ordinal rankings of the FQE value estimates vs. the true values.

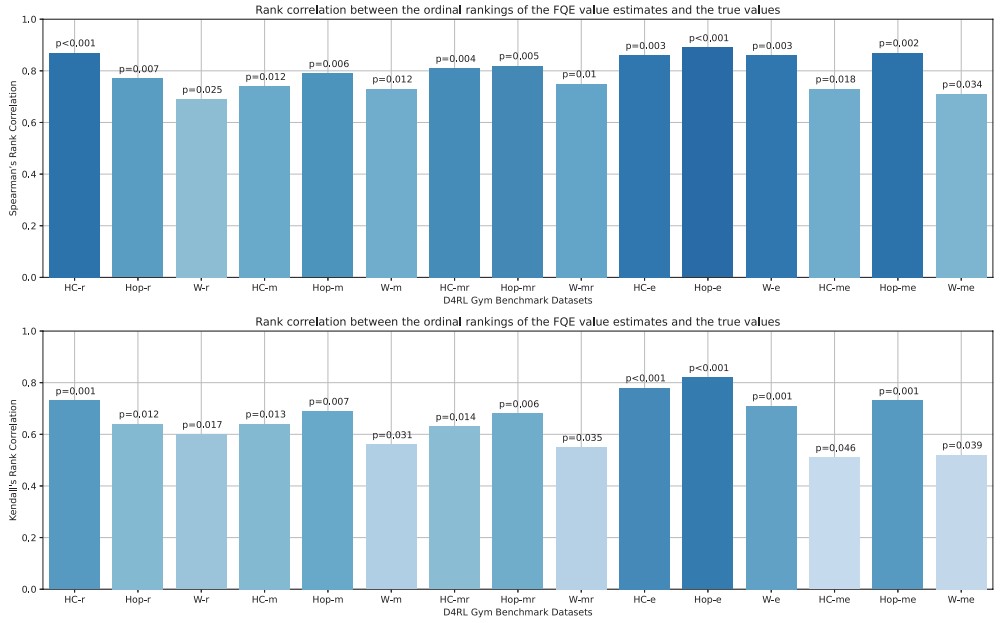

Figure 6: Rank correlation of our FQE algorithm for D4RL Gym benchmark datasets. HC = HalfCheetah, Hop = Hopper, W = Walker, r = random, m = medium, mr = medium-replay, e = expert, me = medium-expert.

## A.8 RESULTS ON THE D4RL GYM EXPERIMENTS

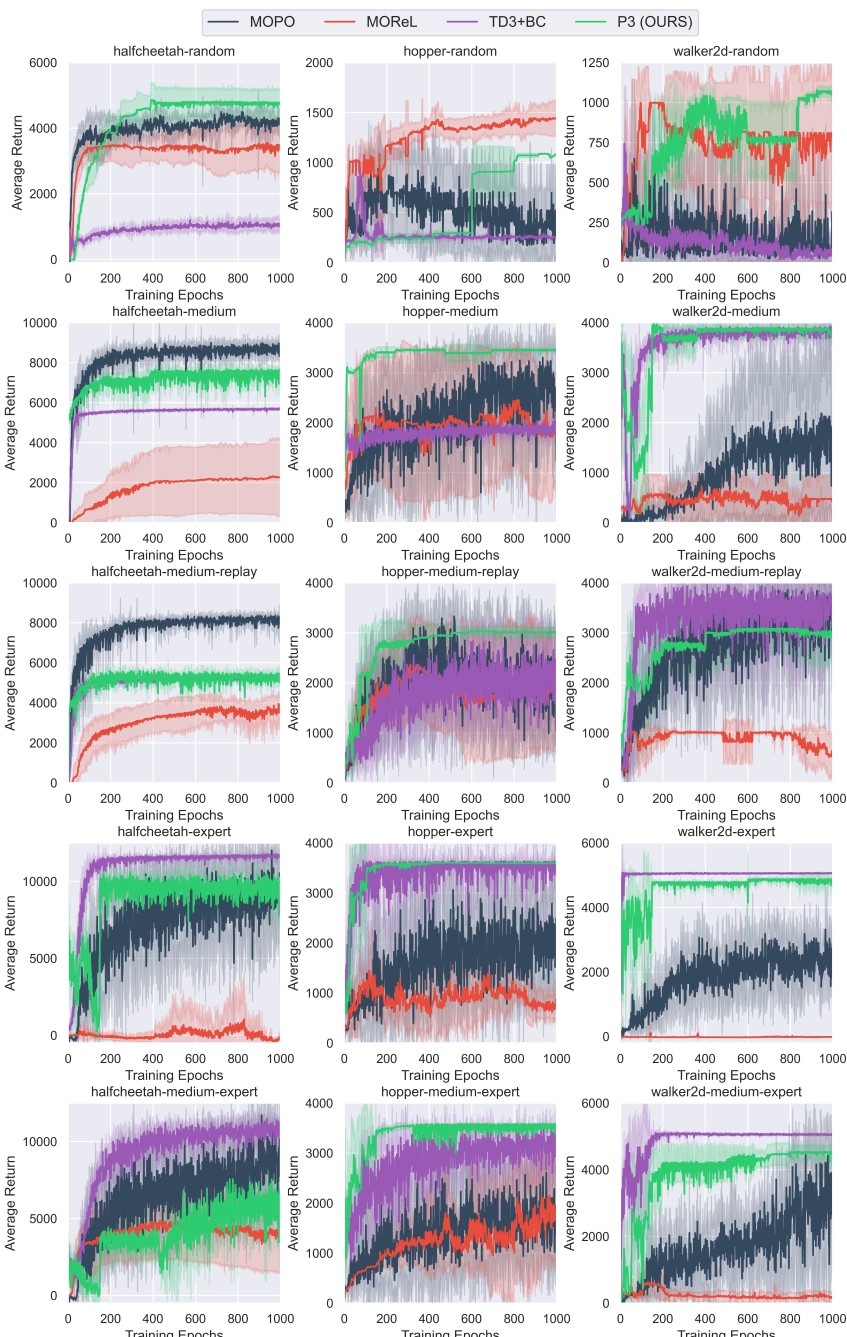

Figure 7: **Learning curves for the D4RL Gym experiments.** Curves are averaged over the 10 evaluations and 5 seeds, and the shaded area represents the standard deviation across seeds. P3 performs better compared to two model-based offline RL methods while exhibiting similar performance as the state-of-the-art model-free method (TD3+BC). Note that we sweep the hyperparameters of MOPO and MOReL and choose the best combination for each task.

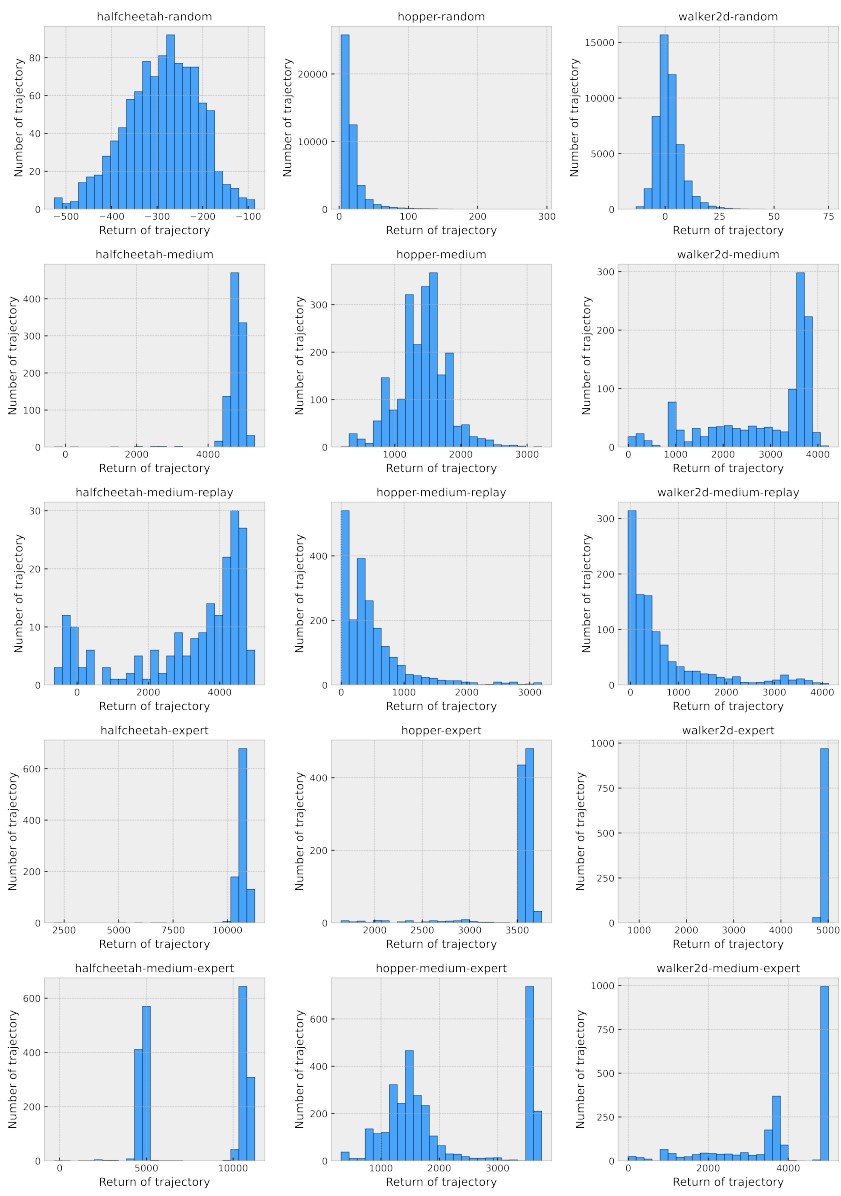

Figure 8: **The histogram of returns from low-quality to high-quality datasets.** Low/medium-quality datasets in the D4RL Gym benchmark are "imbalanced" and degrade the training of environment models. P3 (ours) exhibits more advantages on these datasets.

## A.9 ABLATION STUDY

For clarity in ablation study, we provide implementation details on the scalarization method (Boyd & Vandenberghe, 2004, Chapter 4.7). The method maximizes a weighted sum of objectives: $\max \boldsymbol{\omega}\mathbf{J}(\pi_\theta)$ where $\boldsymbol{\omega} \in [0, 1]$ denotes the weight assignment. By solving multiple single-objective optimization with different weight assignments, we can obtain multiple policies with different trade-offs, as shown in Fig. 9. For the experimental results in Table 2 and Fig. 9, we train 5 policies from scratch using the scalarization method with random weights and report the best score among them.

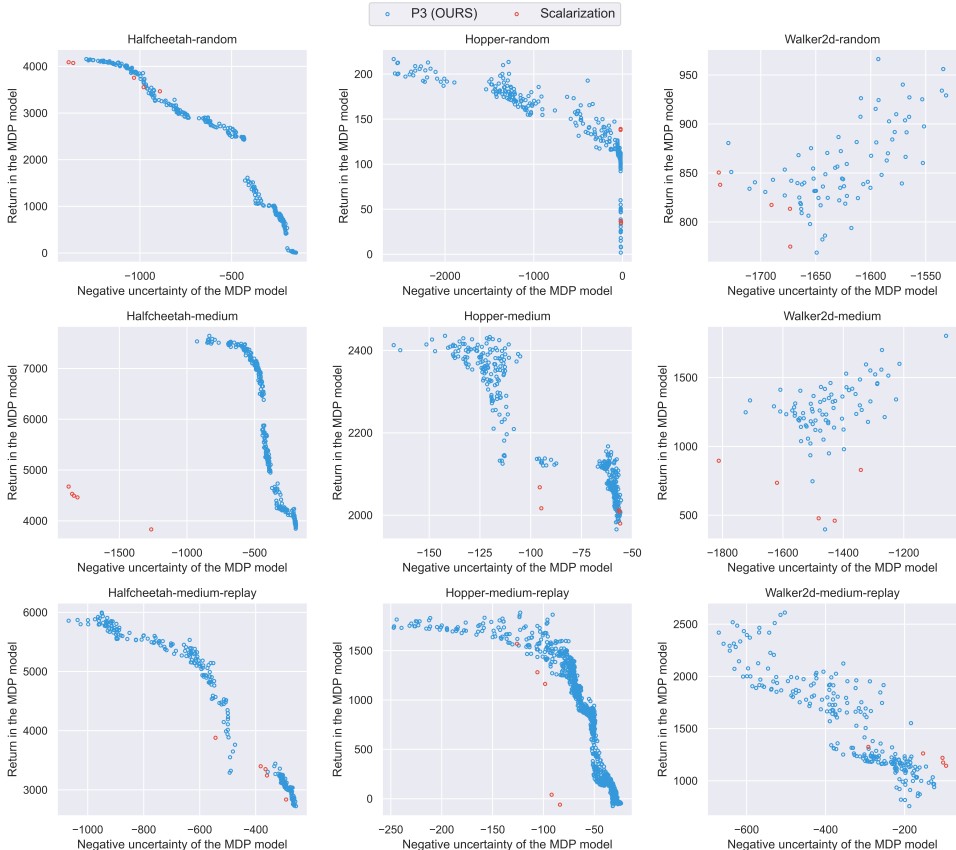

Figure 9: **Ablation study.** Pareto front comparison between the scalarization method (red circles) and our P3 (blue circles) on low/medium-quality datasets. We train 5 policies from scratch using scalarization with random weights. P3 also adopts 5 reference vectors. Hence, we remark that P3 takes the roughly same run time as scalarization but achieves a dense set of policies and better Pareto front approximation.

## A.10 Effectiveness of Pareto policy pool

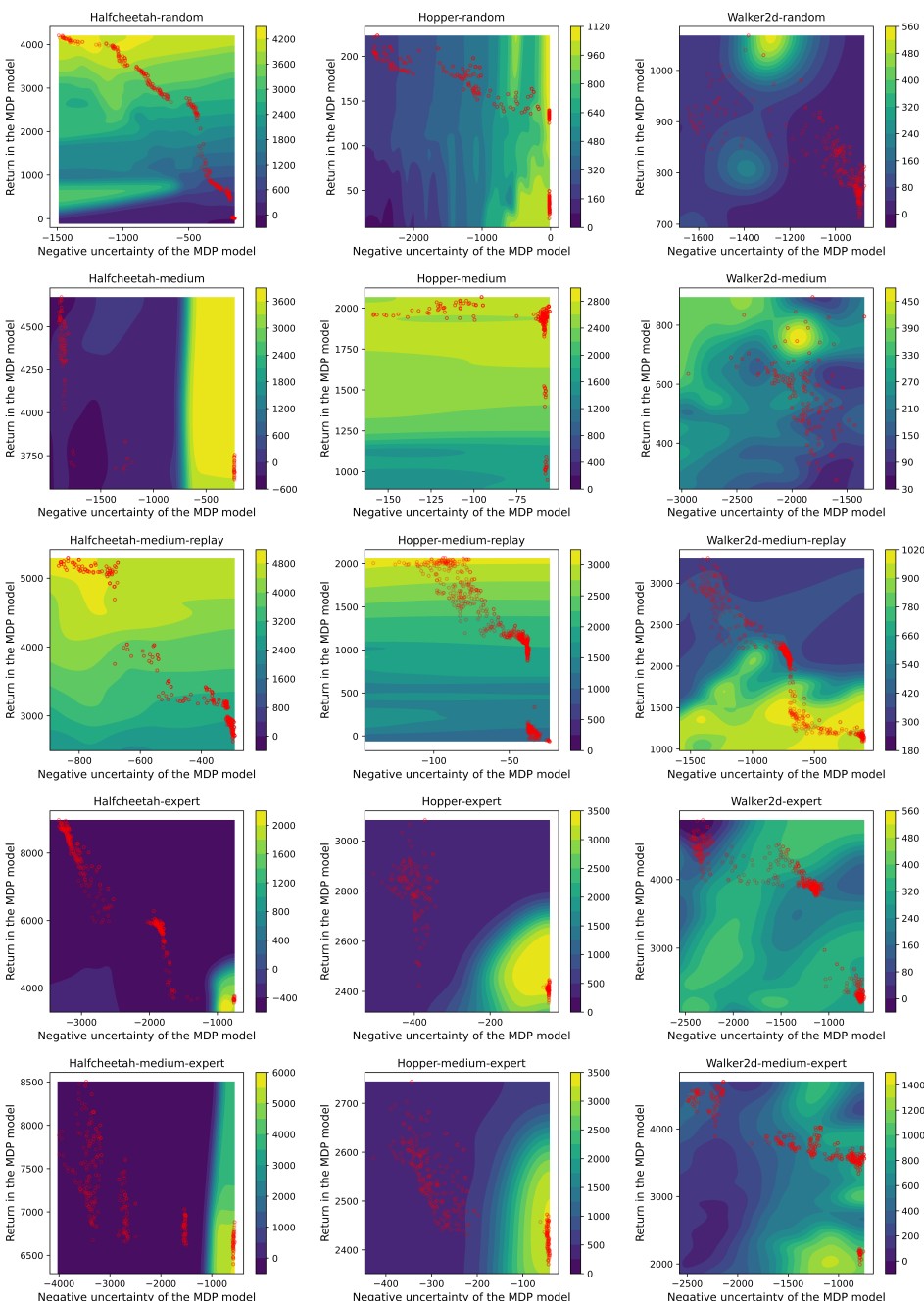

Figure 10: Contour plots on how model-based offline RL's performance (color bar) varies with different trade-offs between the model return (y-axis) and uncertainty (x-axis). Meanwhile, red circles denote the pool of policies generated by P3.

