# OpenReview forum: "Pareto Policy Pool for Model-based Offline Reinforcement Learning"
_ICLR.cc/2022/Conference — ICLR 2022 Poster_

### Official Review · Reviewer_jeW5 · 2021-10-25

**Correctness:** 3
**Technical Novelty And Significance:** 3
**Empirical Novelty And Significance:** 3
**Recommendation:** 8
**Confidence:** 4

**Main Review:**

**Strengths**

I think the paper has the following strengths:
- The paper outlines a novel method, with a thorough and detailed description, for offline model-based reinforcement learning that reformulates a significant problem in a bi-objective manner.
- The paper provides a thorough explanation of the various components of their method, including algorithms, gradients and relevant parameters and hyper parameters.
- The method shows general outperformance compared to other methods in lower data quality settings in the D4RL benchmark.

**Weaknesses**

I think the paper could be strengthened by providing more detail on the following:
- Clarifying how the policies for the score in Table 1 were chosen. Given that you have a Pareto front, did you choose the one with the highest score on the given benchmark? If so, can you provide more detail on how many policies end up in your Pareto front. I am potentially concerned that creating many additional could add significant computation overhead when measuring performance if all policies are tested, so further details could help.
- You mention that you use ES for gradient estimation for the MGDA based optimization. Did you consider other methods and how do you think gradient estimation will impact the quality of solutions along the Pareto frontier that P3 can find?
- Can you provide more detail on how you chose the number of reference vectors for your bi-objective optimization ("n" in Line 5 Algorithm 1). Figure 5 has many (red-dot) Pareto optimal points, so it appears that many reference vectors were chosen. Related to the first question, I am trying to get a sense of the scale of the optimization problem.
- Related question to Figure 5: Based on my assessment it appears that not all red dots representing Pareto policies are non-dominated along the two objective dimensions. Could you clarify if this is the case?
-  Could you comment on why you think UWAC outperforms P3 in the medium-expert and expert cases?

**Additional Remarks**

- Reproducibility: While hyperparameters and settings are provided in the appendix, there is no mention that I can find of code release. Could you provide more detail on whether you plan to release any source code?



**Summary Of The Paper:**

The paper proposes a new model-based offline reinforcement learning, where a pool of Pareto optimal policies is trained on a model of the environment provided by offline RL data. First, a model of the environment is trained using supervised learning from the dataset of logged experiences and subsequently a set of Pareto optimal policies is trained along the dimension of return in the MDP model and uncertainty in the MDP model. The authors argue that providing a Pareto optimal set of policies is superior to prior methods that relied on regularization of the two objectives into a combined metric, as having a Pareto optimal set of policies enables one too explore a range of behavior that is inherent with trade-off between model return vs model uncertainty in offline model based RL.

The authors then describe their method, Pareto Policy Pool (P3), which addresses the bi-objective optimization problem two stage method. Initially, the authors initialize a set of reference vectors along the Pareto frontiers to ensure a diverse set of Pareto policies can be found. Subsequently, the authors apply "Local Pareto Extension" where first a given policy is corrected to a desired range along the objective dimensions provided by the reference vectors. Once the policies are within the desired range, the method applies MGDA based optimization to further improve the performance of the policy in a Pareto optimal way (ascension stage). In order to alleviate computational cost of training multiple policies, the authors apply a "Pareto extension" method to initialize new policies along the Pareto frontier.

The authors then perform experiments on D4RL based benchmarks and compare their method to other literature methods and claim outperformance, particularly in lower quality dataset settings. The authors also perform an ablation study pertaining to the various components of their method, as well as an analysis related to why it is challenging to obtain optimal trade-offs in model based offline RL.

**Summary Of The Review:**

Overall, I think that the paper provides a novel and valuable method with a satisfactory set of experiments to support the overall claims. I have outlined some of my questions and concerns above and think that further clarification will give me a better sense to re-assess my recommendation during the discussion period.

----- Discussion Period Comments -----

I am adjusting my score based on the authors' response to my questions, their modifications in the paper draft and response to the feedback of other reviewers. I will follow up with specific technical questions in the individual response.

---

> ### Author Response · Authors · 2021-11-23
> **Response to Reviewer jeW5**
>
> Thanks for your constructive comments! **In our reply to Q1 of the general response, we addressed your concerns regarding the fairness of the empirical evaluation.** Moreover, we removed the online evaluation of P3 policies during inference for better efficiency and a computation-fair comparison. Instead, we select the best policy by an offline evaluation method “FQE” on the training data. We reported the results of new experiments for P3+FQE in Table 1 and it still outperforms all baseline on low/medium-quality datasets. Here are our detailed replies to your questions.
>
> ### **Q1** *Clarifying how the policies...I am potentially concerned that creating many additional policies could add significant computation overhead…*
>
> Please refer to our reply to Q1 in the general response.
>
> ### **Q2** *Other methods for gradient estimation? ...will impact the quality of solutions...P3 can find?*
>
> **As demonstrated in a number of recent work [1,2] as well as our experiments, ES is a simple yet effective gradient estimator in practice when directly computing gradients is infeasible or computationally prohibitive.** That being said, we also notice some alternatives such as CMA-ES [3], NES [4], and ARS [5]. Theoretically, P3 can use any existing gradient estimator. However, compared to ES, applying these estimators usually require additional hyperparameters and computational cost.
>
> A better gradient estimator can potentially improve the sample efficiency and convergence of Algorithm 2 in P3, resulting in a more diverse set of Pareto policies. **In our experiments, P3 with the simple ES estimator already results in competitive performance on the D4RL benchmark. But we will keep exploring other gradient estimators in the future.**
>
> ### **Q3** *Can you provide more detail on how you chose the number of reference vectors...the scale of the optimization problem.*
>
> **In all experiments of our paper, we only use 5 reference vectors (given in Table 4 of Appendix)** but P3 can still achieve a number of diverse policies covering most parts of the Pareto front, as shown in Fig. 4. This is a primary advantage of P3 which substantially improves the efficiency, thanks to our local Pareto extension strategy detailed in Section 4.2.2 and Line 9-13 in Algorithm 1.
>
> ### **Q4** *Related question to Figure 5...not all red dots representing Pareto policies are non-dominated...clarify if this is the case?*
>
> **As explained in our reply to Q3, only 5 red dots in each plot are guaranteed to be Pareto stationary solutions that are non-dominated**, while the others are found by local Pareto extension (Sec. 4.2.2) based on those 5 dots so they are not guaranteed to be non-dominated. **However, they are sufficiently close to Pareto front, as shown in Fig. 4, and computing them saves a significant amount of computation.**
>
> ### **Q5** *Could you comment on why you think UWAC outperforms P3 in the medium-expert and expert cases?*
>
> Please refer to our reply to Q2 in the general response.
>
> ### **Q6** *Reproducibility.*
>
> We believe that P3 is a general and principal approach that can be integrated with many existing works and bring improvement. **We will definitely release the source code very soon.**
>
> [1] Xingyou Song, Wenbo Gao, Yuxiang Yang, Krzysztof Choromanski, Aldo Pacchiano, Yunhao Tang. ES-MAML: Simple Hessian-Free Meta Learning. ICLR 2020.
> [2] Paul Vicol, Luke Metz, Jascha Sohl-Dickstein. Unbiased Gradient Estimation in Unrolled Computation Graphs with Persistent Evolution Strategies. ICML 2021.
> [3] Anne Auger, Nikolaus Hansen. Tutorial CMA-ES: evolution strategies and covariance matrix adaptation. GECCO, 2013.
> [4] Daan Wierstra, Tom Schaul, Tobias Glasmachers, Yi Sun, Jan Peters, Jürgen Schmidhuber. Natural evolution strategies. J. Mach. Learn. Res. 2014.
> [5] Horia Mania, Aurelia Guy, and Benjamin Recht. Simple random search of static linear policies is competitive for reinforcement learning. In NeurIPS, 2018.

---

> > ### Comment · Reviewer_jeW5 · 2021-11-26
> > **Additional Questions**
> >
> > Thank you for providing additional details and clarifications. I have some follow up questions:
> >
> > **Q1:**
> > * Could you provide more details on why you chose FQE for your evaluation method? Are there other alternatives, if so what are they?
> >
> > **Q3:**
> > * Do you have a sense on what potential trade-offs for choosing higher/lower number of reference vectors would be? Is it mainly a questions of computational expenses or are there other reasons?
> >
> > **Q4:**
> > * It also appears that you are arguing that P3 can find solutions by Pareto extension that are not necessarily Pareto optimal but can still be useful for the algorithm. Can you provide more detail on when you would use a policy that is not Pareto optimal in your overall algorithmic framework?

---

> > > ### Author Response · Authors · 2021-11-28
> > > **Another Response to Reviewer jeW5**
> > >
> > > Thank you for your support and constructive comments! Here are our detailed replies to your follow-up questions.
> > >
> > > ### **Q1** *Could you provide more details on why you chose FQE for your evaluation method? Are there other alternatives, if so what are they?*
> > >
> > > **We chose FQE based on a number of recent works [1-3], which have demonstrated that FQE is more efficient, effective, and simpler to implement than other offline policy evaluation methods.** Specifically, other methods usually require additional hyperparameters and computational cost. Some of them have to additionally solve complicated estimation problems, e.g., learning a dynamic model of high-dimensional observations [4] or estimating the importance weights in a continuous action space [5]. For more comparison and discussion, please refer to a recent survey [2]. It is possible that there might be a better choice of offline policy evaluation method for P3, but the choice might also be task and dataset dependent. Since **P3 is proposed as a general framework that can use any offline evaluation method, we did not focus on the selection of a specific offline evaluation algorithm.**
> > >
> > > ### **Q3** *Do you have a sense on what potential trade-offs for choosing higher/lower number of reference vectors would be? Is it mainly a questions of computational expenses or are there other reasons?*
> > >
> > > Without our local extension, the trade-off between the computational cost of having more reference vectors and the final performance can be significant. However, considering that (1) the optimization of multiple Pareto front policies can be computed in parallel (line 5 of Alg. 1); and (2) our local extension in Section 4.2.2 can find many policies close to the Pareto front more efficiently, **the trade-off does not cause significant difference on the performance.** To demonstrate it, in the table below, we tried P3 with $3,5,10,15$ reference vectors on the halfcheetah-random dataset and it shows that $5$ reference vectors suffice to cover most policies with the local extension and obtain a promising normalized score.
> > >
> > > | Number of reference vectors | 3 | 5 | 10 | 15 |
> > > | :- | :- | :- | :- | :- |
> > > | Normalized score | 30.01 | 38.82 | 40.67 | 39.90 |
> > > | Wallclock time (min) | ~37 | ~37 | ~40 | ~45 |
> > > | Number of compute nodes | 3 | 5 | 10 | 15 |
> > >
> > > ### **Q4** *It also appears that you are arguing that P3 can find solutions...not necessarily Pareto optimal but can still be useful for the algorithm…*
> > >
> > > Solutions achieved by Pareto extension are not necessarily Pareto optimal but can be **sufficiently close to the Pareto front** so they are still diverse and “near-optimal” policies that can provide different levels of model return-uncertainty trade-offs in our policy pool. Selecting the best policy from a large pool including them can increase the chance of getting higher realistic returns.
> > >
> > > **We observed in our ablation study that these near-optimal policies play important roles in P3** and are often selected as the best policy in the test phase. Specifically, in Table 2, when we remove the local extension from P3, i.e., “no ParetoExtension”, P3 selects the best policy only from fewer Pareto policies achieved via reference vectors and its performance on all the 9 datasets substantially degrades. If the near-optimal policies are rarely selected as the best policy during the test, we cannot observe such degradation.
> > >
> > > [1] Tom Le Paine, et al. Hyperparameter selection for offline reinforcement learning. 2020.
> > > [2] Justin Fu, et al. Benchmarks for deep off-policy evaluation. ICLR 2021.
> > > [3] Hoang Minh Le, et al. Batch policy learning under constraints. In ICML, 2019.
> > > [4] Michael R. Zhang, et al. Autoregressive dynamics models for offline policy evaluation and optimization. ICLR 2021.
> > > [5] Mengjiao Yang, et al. Off-policy evaluation via the regularized lagrangian. NeurIPS, 2020.

---

### Official Review · Reviewer_ngyt · 2021-11-02

**Correctness:** 3
**Technical Novelty And Significance:** 2
**Empirical Novelty And Significance:** Not applicable
**Recommendation:** 6
**Confidence:** 3

**Main Review:**

Strengths:
- The paper is well written aside from a few typos here and there. The figures are well made and clearly convey relevant information about the problem, method, and experimental results.
- The problem identified by the authors is highly relevant to the RL and offline RL community. While this problem (and similar ones) has been identified before, there is relatively little prior work that has addressed it despite it being an important problem.
- The proposed method seems technically sound and the techniques for multi-objective optimization in RL seem to be of interest to the community. The theory is not particularly new, but it does validate certain parts of the algorithm, which is nice.
- The experiments are extensive and compare against a large pool of recently proposed methods. P3 appears to very clearly out-perform the other methods when low-medium quality trajectories are given in the dataset. The ablation study and heatmaps are particularly informative to understand exactly what the algorithm is achieving.



Weaknesses:
- The algorithm is purported to apply to the case of offline RL, yet there is a critical concession in practice: one must have online access to the simulator/domain to actually select the best policy from the frontier -- the algorithm is unable to determine this with offline data alone. This is concerning for two reasons. (1) In my opinion, this trivializes the model/policy/hyperparameter selection problem since we can simply try many different hyperparameters for a given algorithm and see which one is best directly, just as one can in standard supervised learning e.g. by cross-validation or hold-out sets. (2) I do not see an easy way to remedy this problem (I am certainly open to and curious about any suggestions) because Figure 5 shows that a large portion of frontier policies still have low reward so random sampling probably won’t work reliably and something like OPE requires hyperparameters of its own.
- Given that some online access is permitted to do the final selection for P3, I think a key experiment that is missing is a comparison with hyperparameter sweeps of the benchmark algorithms like MOReL, MOPO, UWAC, etc. in each domain. That is, if one grids for example the regularization weight $\lambda$ mentioned in Section 2 and picks the best for each domain from online evaluation, how does this compare with P3? Otherwise it seems that P3 has an unfair advantage. I think if such a comparison is made, then the previous issue is not as significant.

Minor comments and questions:
- Equation (4): What is the motivation for the exponentiated uncertainty? How do you set kappa?
- Can you define “non-convex regions” on the Pareto frontier?
- Fig 5 is informative to some extent, but I am confused about how to interpret the full frontier. Clearly there are some points in high reward regions, but also many in seemingly arbitrary areas. Can additional discussion be provided to aid interpretation? Why are some of the points very dense while others are not?
- Some important sections like conclusion and related work seem to be in the appendix.
- For the experiments, how many policies are in the set $\mathcal{P}$ in the end?


**Summary Of The Paper:**

This paper proposes a new algorithm for model-based offline reinforcement learning in which the learner attempts to optimally balance maximizing the reward under the learned model and minimizing the uncertainty of the model. The algorithm is motivated by the fact that offline RL methods suffer from distribution shift that often results in policies/predictions that are overly optimistic about highly uncertain areas of the state-action space. The authors’ proposed method attempts to address this problem by identifying a relevant collection of policies lying on the Pareto frontier that trades off these two criteria. The underlying supposition is that a well-performing policy should be among this collection of “non-dominated” policies on the frontier. Specialized subroutines are proposed in order to make sure the collection is diverse. Some theory is presented to validate certain aspects. Extensive experimental results support that the method performs well in relevant benchmarks compared to a number of state-of-the-art methods.


**Summary Of The Review:**

In summary, I believe there are a lot of positives about this paper. It identifies and studies an important problem in the literature. The technical contributions and main ideas are interesting and appear to be new to this problem setting. The experiments are reasonably thorough. Unfortunately, I feel that the mentioned weaknesses (requiring online access, lack of comparison against hyperparameter sweeping other algorithms) are significant and leave open questions about the practical applicability of the proposed method and comparison to prior algorithms.

---

> ### Author Response · Authors · 2021-11-23
> **Response to Reviewer ngyt (Part 1/2)**
>
> Thanks for your constructive comments! **In our reply to Q1 of the general response, we addressed your concerns regarding the fairness of the empirical evaluation.** Moreover, we removed the online evaluation of P3 policies during inference for better efficiency and a computation-fair comparison. Instead, we select the best policy by an offline evaluation method “FQE” on the training data. We reported the results of new experiments for P3+FQE in Table 1 and it still outperforms all baseline on low/medium-quality datasets. Here are our detailed replies to your questions.
>
> ### **Q1** *The algorithm is purported to apply to the case of offline RL, yet there is a critical concession in practice…*
>
> Please refer to our reply to Q1 in the general response.
>
> ### **Q2** *Need to tune hyperparameters for other baselines.*
>
> **In our experiments, for fair comparisons, we carefully tuned the hyperparameters of baselines such as BCQ, CQL, MOPO, and MOReL by grid search and chose the one achieving the best performance on each benchmark dataset.** Reviewer xWzd noticed our efforts on tuning the baselines and here is a quote from her/his comment: "I also appreciate that the authors have taken efforts to tune the MOPO baseline, in some cases even finding hyperparameters that work better than the original paper's results." In addition to this, our results of BCQ and TD3+BC are also better than those reported in the original papers [1,2]. For other baselines such as UWAC and COMBO, we adopt the hyperparameters they use in their original papers and we presume that they have carefully tuned the hyperparameters to achieve the best performance.
>
> ### **Q3** *Equation (4): What is the motivation for the exponentiated uncertainty? How do you set kappa?*
>
> Please refer to our reply to Q3 in the general response.
>
> ### **Q4** *Can you define “non-convex regions” on the Pareto frontier?*
>
> **Non-convex regions of a Pareto front refer to the connected non-convex sets of Pareto stationary solutions on the Pareto front.** A non-convex set $S$ is defined as: Given arbitrary $x$ and $y$ in $S$, and $t$ in the interval $[0,1]$, a set $S$ is non-convex if there exists $(1-t)x+ty$ not belonging to $S$.
>
> ### **Q5** *...Clearly there are some points in high reward regions, but also many in seemingly arbitrary areas...additional discussion be provided to aid interpretation?*
>
> This is a result of the well-known gap between the realistic environment and the MDP models learned in model-based RL: the heatmap reflects the realistic return of the former, while the Pareto front is for the latter. **Hence, not all Pareto front solutions perform well in the realistic environment (only those located at the high-reward regions of the heatmap).**
>
> **This gap actually motivates the development of P3:** since we do not have access to the realistic environment and the heatmap during offline RL, solely optimizing either the model return, the model uncertainty, or their weighted sum may result in sub-optimal policies performing poorly in terms of the realistic return (i.e., the heatmap). **By exploring the whole Pareto front in an efficient manner, P3 is able to find more and diverse Pareto policies that have more chances to achieve higher realistic returns.**
>
> [1] Justin Fu, Aviral Kumar, Ofir Nachum, George Tucker, and Sergey Levine. D4rl: Datasets for deep data-driven reinforcement learning. 2020.
> [2] Scott Fujimoto and Shixiang Shane Gu. A Minimalist Approach to Offline Reinforcement Learning. 2021.

---

> > ### Author Response · Authors · 2021-11-23
> > **Response to Reviewer ngyt (Part 2/2)**
> >
> > ### **Q6** *Why are some of the points very dense while others are not?*
> >
> > **In practical multi-objective optimization problems, the solutions are not uniformly distributed over the Pareto front, so the density of solutions varies across different regions. This is commonly observed in previous studies [3-5] and is one of the major challenges for P3.** In our results, even though the density varies drastically for different regions, P3 can still find a certain number of solutions in the low-density regions. This is important to the final improvement observed because the achieved policy pool covers policies that can perform diverse levels of trade-off between the two objectives, hence resulting in more chances to find better policy during inference.
> >
> > ### **Q7** *Some important sections like conclusion and related work seem to be in the appendix.*
> >
> > Thanks for the suggestion! In the new version, we moved the related works and the conclusion section to the main paper.
> >
> > ### **Q8** *For the experiments, how many policies are in the set P in the end?*
> >
> > As illustrated in Algorithm 1, if $n=5$, $T_l=25$, we will obtain $5\times2\times25$ policies in the set $\mathcal{P}$.
> >
> > [3] Jie Xu, et al. Prediction-guided multi-objective reinforcement learning for continuous robot control. In ICML, 2020.
> > [4] Adriana Schulz, et al. Interactive exploration of design trade-offs. ACM Trans. Graph. 2018.
> > [5] Jie Xu,et al. Multi-objective graph heuristic search for terrestrial robot design. ICRA 2021.

---

> > > ### Comment · Reviewer_ngyt · 2021-11-29
> > > **Thanks**
> > >
> > > Thanks for your detailed response to the review. I appreciate the additional comparisons with P3+FQE in Table 1. I’ve raised my score for now. However, I have a few remaining comments. While the correlation experiments are useful for getting a rough idea of how similar the results might be to the original P3, I think the case for this method would be a lot stronger if the remaining experiments (learning curves, ablations, etc) and description of the method itself would focus solely on the P3+FQE method since this is the only truly offline version. In my view, the original P3 is more of an “oracle” version that is not something that could really be used in practice.
> > >
> > > Also, as mentioned in the initial review, the addition of an offline policy evaluation method seems to introduce additional (potentially difficult to tune) hyperparameters, which can reintroduce the chicken-and-egg problem similar to what this paper is trying to address from the beginning. While not quite a deal breaker, this seems important to address perhaps in some discussion since a critical step in P3 is evaluation of the pool of policies. I am also curious to hear further discussion of Q1 from Reviewer EL3w.

---

> > > > ### Author Response · Authors · 2021-11-30
> > > > **Response to the Remaining Concerns of Reviewer ngyt**
> > > >
> > > > Thank you for your support and constructive comments! Here are our detailed replies to your remaining concerns.
> > > >
> > > > ### *While the correlation experiments are useful for getting a rough idea...this method would be a lot stronger if the remaining experiments (learning curves, ablations, etc)...focus solely on the P3+FQE method*
> > > >
> > > > **Table 1 reports our main and final results on all the standard benchmark datasets** commonly used in model-based offline RL and it suffices to demonstrate P3+FQE’s efficiency and effectiveness over all datasets, especially on those low/medium-quality ones. All the other plots aim at showing the intermediate results or empirical analysis for the experiments in Table 1. **We will add the results of P3+FQE to the remaining plots and tables, as you suggested.**
> > > >
> > > > ### *Also...seems to introduce additional (potentially difficult to tune) hyperparameters, which can reintroduce the chicken-and-egg problem ...this seems important to address perhaps in some discussion since a critical step in P3 is evaluation of the pool of policies.*
> > > >
> > > > - **We did not try other offline policy evaluation methods** and **we did not heavily tune the hyperparameters of FQE** but it already produces promising results and a more efficient P3 variant in Table 1 when combined with P3.
> > > >
> > > > - **A key reason for choosing FQE here is that FQE only has a few hyperparameters** ($K_{epoch}$, $\gamma$, $\beta$), where $\gamma$ and $\beta$ are usually set to default values for MuJoCo tasks without tuning according to previous works [1-3], **so only $K_{epoch}$ needs to be tuned**. In our experiments, we simply use the implementation by [4] and all its default hyperparameters. As demonstrated by [4], FQE is not very sensitive to the choice of epochs $K_{epoch}$ when it is sufficiently large. In our experiments, we did not tune it for every dataset but found  $K_{epoch}=500$ is sufficient to find a near-optimal policy.
> > > >
> > > > - A recently proposed variant of FQE, i.e., “FQE+BVFT” (<https://arxiv.org/pdf/2110.14000.pdf>) is designed to be hyperparameter-free. **By using it, we may entirely remove the risk of the chicken-and-egg problem.** However, this is still an open problem and is not the current focus of our paper. We will keep trying other offline evaluation methods to find a better solution for it.
> > > >
> > > > ### *I am also curious to hear further discussion of Q1 from Reviewer EL3w.*
> > > >
> > > > For more detailed discussions, please refer to our new reply to Q1 from Reviewer EL3w.
> > > >
> > > > [1] Achiam, Joshua. Spinning up in deep reinforcement learning. 2018.
> > > > [2] Tuomas Haarnoja, et al. Soft actor-critic algorithms and applications. 2018.
> > > > [3] Stable baselines3: <https://github.com/DLR-RM/stable-baselines3>.
> > > > [4] Tom Le Paine, et al. Hyperparameter selection for offline reinforcement learning. 2020.

---

### Official Review · Reviewer_EL3w · 2021-11-02

**Correctness:** 3
**Technical Novelty And Significance:** 3
**Empirical Novelty And Significance:** 3
**Recommendation:** 5
**Confidence:** 3

**Main Review:**

In terms of strengths, this paper presents a novel perspective on navigating the trade-off between model-based return versus uncertainty, for offline RL. The experiments are thorough in comparing against a range of baselines, including both model-based and non-model-based offline RL algorithms. Figures 1 and 2 do a nice job of visualizing the approach and the results.

In terms of weaknesses, a main weakness is that P3 does not outperform existing approaches for good-quality offline RL datasets: in Table 1, for the medium-expert and expert datasets, it consistently performs substantially worse then the UWAC baseline. Why is this the case? I would expect P3 to perform at least as well, if it is actually recovering the true Pareto front. Another weakness is that the experiments are on a relatively small number of domains—only on three MuJoCo domains, out of the many domains available in D4RL.

Also, in practice, trying out every single (fully-trained) policy in the real environment is still expensive. This should be acknowledged in the paper. If I understand correctly, the results in Table 1 for P3 show the best performance across all policies found. This is significantly more computationally expensive than prior approaches (because at least several of the Pareto policies require separate training from scratch), and also requires more runs on the real environment. So it is not entirely a fair comparison. With that said, I appreciate the ablation studies that make a more fair comparison against ablations of P3.

Finally, the algorithmic design decisions should be motivated more clearly. In particular, when computing the similarity metric between the reference vector and the objective vector in Equation (6), why is KL-divergence used instead of a dot product? The dot product is the typical way of measuring similarity between vectors (as is the case here), whereas KL-divergence is meant to be used for distributions. Also, in Equation (4), does the uncertainty need to be exponentiated and tempered?

Additional comments:
- The Related Work and Conclusion should be in the main paper, not in the Appendix.
- In the Related Work, it would be useful to also discuss non-model-based approaches for offline RL. It is also worth mentioning DiME [1], a recent approach that, similar to P3, also takes a multi-objective perspective for offline RL (although in a non-model-based setting), and obtains state-of-the-art results by doing so.
- There are several typos and grammatical errors. e.g. "computational forbidden" at the bottom of page 3
- In the last paragraph of Section 3.1, the writing is overly verbose and unclear. For instance, it mentions twice that it is inefficient to train multiple policies from scratch to obtain the Pareto front.
- Section 4.2 claims that scalarization does not perform well because it only finds policies on the convex part of the Pareto front, but the Pareto fronts found by P3 (in Figure 8) don’t look concave. Instead, I think the different scales of the objectives is the issue, because only five weight settings for linear scalarization are tried.
- In A.8, for walker2d-random and walker2d-medium, why do these not look like Pareto fronts? In other words, why is there not a conflict between optimizing for model-based return and minimizing model uncertainty for this task?

[1] Abdolmaleki et al. On Multi-objective Policy Optimization as a Tool for Reinforcement Learning. 2021.

**Summary Of The Paper:**

This paper proposes a model-based approach for offline RL that is inspired by multi-objective RL. The approach, called P3, finds a Pareto front of policies that trade off between obtaining high return (with respect to the model trained on the offline dataset) and minimizing the model's uncertainty. To obtain this Pareto front, the approach has two stages: 1) find a few spread-out solutions on the Pareto front using a gradient-based method, and 2) initialize with these solutions to find optimal solutions for the neighboring regions of the Pareto front. The empirical evaluation shows that P3 outperforms existing approaches when the offline dataset is of low or medium quality.

**Summary Of The Review:**

This is an interesting and relevant direction to study, with promising empirical results for tasks with low- and medium-quality datasets. However, the algorithmic design decisions and empirical results (for good-quality datasets) require more explanation, and the presentation requires polishing.

---

> ### Author Response · Authors · 2021-11-23
> **Response to Reviewer EL3w (Part 1/2)**
>
> Thank you for your constructive comments! **In our reply to Q2 of the general response, we addressed your concerns regarding the explanation of empirical results. In our reply to Q1 of the general response, we addressed your concerns about the fairness of the empirical evaluation.** Moreover, we removed the online evaluation of P3 policies during inference for better efficiency and a computation-fair comparison. Instead, we select the best policy by an offline evaluation method “FQE” on the training data. We reported the results of new experiments for P3+FQE in Table 1 and it still outperforms all baseline on low/medium-quality datasets. Here are our detailed replies to your questions.
>
> ### **Q1** *A main weakness is that P3 consistently performs substantially worse than the UWAC baseline. Why is this the case?*
>
> Please refer to our reply to Q2 in the general response.
>
> ### **Q2** *Another weakness is that the experiments are only on three MuJoCo domains, out of the many domains available in D4RL.*
>
> **Most of the existing methods in the area, e.g., MOPO and TD3+BC, chose the three MuJoCo domains in D4RL for comparison in their papers, and we followed their settings.** These tasks are sufficiently challenging and representative for a thorough comparison. This is acknowledged by other reviewers and here is a quote from Reviewer ngyt: "The experiments are extensive and compare against a large pool of recently proposed methods." That being said, we will keep exploring the possibility of applying P3 to more different domains, and we believe that P3 and its idea of learning a diverse pool of policies provides a general and principal solution to a variety of problems.
>
> ### **Q3** *Also, in practice, trying out every single (fully-trained) policy...also requires more runs on the real environment...is not entirely a fair comparison.*
>
> Please refer to our reply to Q1 in the general response.
>
> ### **Q4** *This is significantly more computationally expensive than prior approaches (because at least several of the Pareto policies require separate training from scratch)...not entirely a fair comparison.*
>
> At first glance, P3 may cost more computation than prior methods on finding multiple policies. **However, in practice, the computational cost of running P3 is comparable to that of other single-policy methods (e.g., MOPO and MOReL)**, because they need to carefully tune and try different regularization weights. Instead, the multi-objective optimization in P3 does not need to tune the weight.
>
> **In our experiments, for fair comparisons, we carefully tuned the hyperparameters of baselines such as BCQ, CQL, MOPO, and MOReL by grid search and chose the one achieving the best performance on each benchmark dataset.** Reviewer xWzd noticed our efforts on tuning the baselines and here is a quote from her/his comment: "I also appreciate that the authors have taken efforts to tune the MOPO baseline, in some cases even finding hyperparameters that work better than the original paper's results." For other baselines such as UWAC and COMBO, we adopt the hyperparameters they use in their original papers and we presume that they have carefully tuned the hyperparameters to achieve the best performance.
>
> ### **Q5** *...Equation (6), why is KL-divergence used instead of a dot product?*
>
> **KL-divergence and dot-product (as similarity metrics) are the same to our method** in this paper **because we use the rank of similarity to compute the gradient** (as detailed in Appendix A.6) when using OpenAI’s evolution strategy [1]: KL divergence-based similarity and dot-product similarity result in the same rank and thus the same estimate of gradients.
>
> ### **Q6** *Exponential and temperature applied to the uncertainty in Eq. (4).*
>
> Please refer to our reply to Q3 in the general response.
>
> [1] Tim Salimans, et al. Evolution strategies as a scalable alternative to reinforcement learning. 2017.

---

> > ### Author Response · Authors · 2021-11-23
> > **Response to Reviewer EL3w (Part 2/2)**
> >
> > ### **Q7** *The Related Work and Conclusion should be in the main paper, not in the Appendix.*
> >
> > Thanks for the suggestion! In the new version, we moved the related works and the conclusion section to the main paper.
> >
> > ### **Q8** *In the Related Work, it would be useful to also discuss non-model-based approaches for offline RL. It is also worth mentioning DiME.*
> >
> > Thanks for the suggestion! **In the new version, we added discussions to the model-free methods for offline RL such as the concurrent work DiME**. Here is a summary of the differences between P3 and DiME.
> >
> > DiME trains multiple policies from scratch using a modified scalarization method, which cannot cover the non-concave parts of the Pareto front. In contrast, P3 can cover more diverse policies on both the concave and non-concave parts via different reference vectors. Moreover, our local Pareto extension strategy avoids training every policy from scratch and is more efficient than DiME.
> >
> > ### **Q9** *typos and grammatical errors*
> >
> > Thanks for the suggestion. In the new version, we have removed the typos and grammatical errors and we will keep improving the paper.
> >
> > ### **Q10** *In the last paragraph of Section 3.1, the writing is overly verbose and unclear.*
> >
> > In the new version, we carefully rewrote the last paragraph of Section 4.1 and polished other parts of paper according to all reviewers' comments.
> >
> > ### **Q11** *Section 4.2 claims that scalarization does not perform well...the Pareto fronts found by P3 (in Figure 8) don’t look concave...*
> >
> > **In most practical scenarios (and in Figure 8), the Pareto front is neither convex nor concave but is a mix of both, i.e., it has some convex parts and some concave parts [2]**. While scalarization can explore the convex parts, it cannot find a single solution on the concave parts. **This is a well known result of scalarization [3, Chapter 4.7].** In our case, this means that the policy pool achieved by **scalarization CANNOT cover many Pareto policies from the concave parts** that can potentially perform the best on some realistic environments during inference. So its performance can be sub-optimal and much poorer than P3.
> >
> > ### **Q12** *In A.8...why do these not look like Pareto fronts? In other words, why is there not a conflict...for this task?*
> >
> > **Since most solutions achieved by scalarization gather in a small region of the Pareto front (which is another noticeable drawback of scalarization), we only show this small region of the Pareto front in Fig. 9.** The full Pareto front can be found in Fig. 10, which clearly shows a conflict between optimizing for model-based return and minimizing model uncertainty.
> >
> > [2] Deb, K. (2008). Multi-objective optimization using evolutionary algorithms.
> > [3] Stephen Boyd and Lieven Vandenberghe. (2004). Convex Optimization.

---

> > > ### Comment · Reviewer_EL3w · 2021-11-29
> > > **Thank you for the response**
> > >
> > > I appreciate the authors' detailed reply. It addresses most of my concerns, and I appreciate the additional experiments that combine P3 with FQE. I also appreciate the editing and rewriting of the confusing parts of the paper. My main concern regarding the worse performance on high-quality datasets is not addressed though, so I will keep my original score.
> > >
> > > Q1: The reply does not address why P3 performs *worse* than the UWAC baseline for high-quality datasets. This implies that P3 cannot find the best policies in the low uncertainty + high return region. I don't understand why P3 doesn't find these policies, because policies in the low uncertainty + high return region would dominate those that are high uncertainty + high return and those that are low uncertainty + low return, so they should be easily found. In other words, I understand that exploring the whole Pareto front would benefit more when the datasets are low- or medium-quality. But I don't understand why exploring the whole Pareto front would *hurt* when the dataset is high quality, because the best policies (that are low uncertainty + high return) are still part of this Pareto front.
> > >
> > > Q8: Actually, DiME can find points on the non-concave parts of the Pareto front - there are experiments in the paper that demonstrate this empirically. It's not doing scalarization of rewards, but rather combining action distributions. There are also experiments that use DiME to learn the entire Pareto front in a single training run, instead of training every policy from scratch.
> > >
> > > Q11: I see that there are some concave parts of the Pareto front in Figure 8 (now Figure 9). But I still think part of the issue is that not enough linear scalarization weights are tested. There are clearly convex parts of the Pareto front that linear scalarization also does not find solutions on (because only five weights are tested).

---

> > > > ### Author Response · Authors · 2021-11-30
> > > > **Response to the Remaining Concerns of Reviewer EL3w**
> > > >
> > > > Thanks for the response! Here are our detailed replies to your remaining  concerns.
> > > >
> > > > ### **Q1** *The reply does not address why P3 performs worse than the UWAC baseline for high-quality datasets...*
> > > >
> > > > - The reason is that we compared P3 with UWAC (and all baselines) using the same number of gradient updates (i.e., 1000). Here, **UWAC uses 1000 updates to optimize one single policy in the region, while P3 uses 1000 updates to explore many policies over the whole Pareto front.** Therefore, when the optimal policy is known to be allocated in low uncertainty + high return region (for the high-quality dataset), UWAC can use much more gradient updates to refine this region’s solution than P3 and thus performs better. We believe (and in theory it does hold) that P3 can achieve the same performance of UWAC on high-quality datasets, if provided with more computational budget.
> > > >
> > > > - In fact, on high-quality datasets, because we already know which region the optimal policy resides in, it is less necessary to explore the whole Pareto front using P3, and UWAC might be a more efficient solution. **P3 focuses on low/medium-quality datasets which are more common in practice but are more challenging to existing methods.**
> > > >
> > > > ### **Q8** *Actually, DiME can find points on the non-concave parts of the Pareto front...DiME to learn the entire Pareto front in a single training run...*
> > > >
> > > > - **DiME cannot guarantee finding points on the non-concave parts of the Pareto front**, though it performs linear scalarization in another space (action distributions). We cannot find any relevant analysis (except the plots) in the DiME paper: the DiME paper only shows some empirical results of DiME finding points on non-concave parts but they can be dataset dependent without theoretical guarantee. In the multi-objective optimization community, it is well-known that linear scalarization cannot guarantee to find solutions on the non-concave parts of Pareto front (for maximization problems).
> > > >
> > > > - In addition, **a “single training run” in DiME is much more computationally expensive than multiple training runs in P3** because DiME targets a much more ambitious yet difficult problem (similar to [1]) than P3: “a single training run” in DiME learns a meta-model/hyper-network that can directly output a policy of any preferred trade-off level (as the input of the meta-model). In order words, DiME aims at learning a mapping from a low-dimensional reference vector to its associated solution (policy model) on the Pareto front. Instead of finding only 5 Pareto front points in P3, DiME aims at finding all Pareto front points! In order to learn an accurate mapping, DiME has to explore various possible trade-off levels and train the meta-model to produce a policy for each trade-off level: this can be computationally prohibitive in practice for complicated tasks.
> > > >
> > > > - According to <https://iclr.cc/Conferences/2022/ReviewerGuide>, DiME is a concurrent and non peer-reviewed work (it was submitted to arXiv on June 15, 2021, after June 5, 2021).
> > > >
> > > > ### **Q11** *...But I still think part of the issue is that not enough linear scalarization weights are tested...*
> > > >
> > > > - To achieve the results in Fig. 9, P3 only finds 5 Pareto front points by MGDA and the rest points are obtained by cheap local extension. So the computational cost is similar to that of linear scalarization which also finds 5 Pareto front points in Fig. 9. However, the 5 points of linear scalarization are not as diverse as the 5 points achieved by P3: we can see in Fig. 9 that some red points of linear scalarization gather together in a tiny region while the points found by P3 better cover the whole Pareto front because the 5 points found by MGDA are associated with 5 evenly separated reference vectors as in Fig. 2 (we will highlight the 5 points of P3 in Fig. 9 in the next version).
> > > >
> > > > - Moreover, it is well known in multi-objective optimization that **the solutions found by linear scalarization cannot cover a single point in non-concave regions of the Pareto front, no matter how many different weights are tried**. It has also been empirically observed that the solutions of linear scalarization often lack diversity and some easily  converge to the same point. Hence, even equipped with our Pareto extension strategy, the scalarization method cannot provide sufficiently diverse Pareto policies, e.g., $[\pi_i]_{i=1}^4$ in Fig. 2.
> > > >
> > > > [1] Aviv Navon, Aviv Shamsian, Ethan Fetaya, Gal Chechik: Learning the Pareto front with hypernetworks. ICLR 2021.

---

### Official Review · Reviewer_xWzd · 2021-11-04

**Correctness:** 4
**Technical Novelty And Significance:** 3
**Empirical Novelty And Significance:** 3
**Recommendation:** 8
**Confidence:** 4

**Main Review:**

The paper is well-motivated, and the idea to approximate the Pareto front between model-estimated performance and uncertainty is novel to my knowledge. The algorithmic choices (e.g. local extension to reduce computational cost of training from scratch) also make sense. Algorithm 2 is intuitive, although not particularly novel as it is essentially just applying MGDA with gradients estimated by ES, plus some small modifications to account for the constraint.

The performance of P3 on the D4RL tasks is good. UWAC is clearly better for cases where there is expert data, but for the lower-quality datasets, P3 handily outperforms the baselines. I also appreciate that the authors have taken efforts to tune the MOPO baseline, in some cases even finding hyperparameters that work better than the original paper’s results.

My main concern is regarding the fairness of the empirical evaluation, where P3 arrives with a large set of policies to be evaluated, unlike the other methods which produce only one policy (or one for each hyperparameter setting). The authors explain that the computational cost of running P3 is similar to the cost of tuning other methods, which I can believe, but even if you don’t tune P3’s hyperparameters, you still have to draw samples to evaluate each policy, and in the offline setting we do not want to be drawing so many additional samples. Thus, while the authors frame the development of many diverse policies as a strength of their method, I feel that it is only a strength when paired with a strategy to choose from this set of policies (i.e. off-policy evaluation), which is not presented in this paper. At the very least, I feel that the authors should control for the number of samples used at model selection time when comparing against prior work, as this is arguably more of a bottleneck than runtime in offline RL. Currently P3 has an advantage at evaluation time because you get to try many policies and take the max.

**Summary Of The Paper:**

In model-based offline reinforcement learning, a dynamics model is trained from the dataset and subsequently used to produce a decision-making policy which attains high reward (according to the model). However, one must also take care not to allow the policy to exploit the model’s inaccuracies, so we also want it to visit states and take actions which have low model uncertainty. Prior work has managed this tradeoff by linearly combining the two objectives.

This work proposes instead to find many diverse policies along the Pareto front of these two objectives. The algorithm, Pareto policy pool (P3), proceeds as follows:
1. Generate reference vectors which quantify tradeoffs in the Pareto front.
2. In parallel, find a Pareto-optimal policies for each reference vector using “Algorithm 2” to solve a constrained bi-objective optimization problem, where the constraint ties the solution to the reference vector. Algorithm 2 begins by finding a feasible solution and then applies MGDA to improve the objectives.
3. “Local extension”: Each reference vector is perturbed in two opposing directions, and then each perturbed vector is further optimized via Algorithm 2, with intermediate policies being added to the policy pool. (This is just an optimization to reduce the computational cost.)
4. At test time, each policy in the pool is evaluated, and the best one is selected.

The paper includes some theoretical analysis of Algorithm 2, showing convergence to an approximate stationary point. Empirical evaluation is performed on the D4RL benchmark, where P3 achieves good results, particularly on the datasets generated by lower-performing policies.

**Summary Of The Review:**

I think the high-level strategy pursued by the paper is reasonable, and the results are promising. Thus, the paper is a useful contribution overall. However, I have reservations about the setting of the empirical evaluation, and think more care could be taken to ensure a fair comparison.

---

> ### Author Response · Authors · 2021-11-23
> **Response to Reviewer xWzd**
>
> Thank you for your comments! In our reply to Q1 of the general response, we addressed your concern regarding the fairness of the empirical evaluation. Moreover, **we removed the online evaluation of P3 policies during inference for better efficiency and a computation-fair comparison**. Instead, we select the best policy by an offline evaluation method “FQE” on the training data. We reported the results of new experiments for P3+FQE in Table 1 and it still outperforms all baseline on low/medium-quality datasets.

---

### Author Response · Authors · 2021-11-23
**Response to All Reviewers**

We would like to appreciate every reviewer for your time and constructive comments!  We appreciate that reviewers acknowledge that our work is well-motivated (Reviewer xWzd), well-written (Reviewer ngyt), novel (Reviewer xWzd, EL3w, ngyt, and jeW5), and provides thorough experiment results (Reviewer xWzd, ngyt, and jeW5). We also notice that there exist some common concerns on the fairness of empirical evaluation (**Q1**), the explanation of empirical results (**Q2**), and the motivation of algorithmic design (**Q3**), which we will address in the following:

### **Q1** (Reviewer xWzd, EL3w, ngyt, and jeW5) *Evaluating each Pareto policy in the real environment is still expensive, resulting in unfair comparison to other baseline methods.*

We agree that the online evaluation of multiple policies required by the inference for P3 can be expensive. A more efficient solution is to replace the online evaluation with **offline policy evaluation using only the training (offline) data**, which results in **the same inference cost as other baselines** and thus provides a more fair comparison.

Although offline policy evaluation/selection is still an open problem [1-4] out of the scope of this paper, in Appendix A.4 of the updated version, we studied to use an offline evaluation method “Fitted Q Evaluation (FQE)” for the inference of P3. **We report the experimental results of P3+FQE in Table 1, in which “P3+FQE” still outperforms all baselines on all low/medium-quality datasets.** Moreover, in Appendix A.7, we observe a strong correlation between FQE and online policy evaluation results, which indicates that offline policy evaluation enables more efficient inference for P3 and results in a computation-fair comparison.

[1] Sergey Levine, et al. Offline reinforcement learning: Tutorial, review, and perspectives on open problems. 2020.
[2] Cameron Voloshin, et al. Empirical study of off-policy policy evaluation for reinforcement learning. 2019.
[3] Tom Le Paine, et al. Hyperparameter selection for offline reinforcement learning. 2020.
[4] Justin Fu, et al. Benchmarks for deep off-policy evaluation. ICLR 2021.

### **Q2** (Reviewer EL3w and jeW5) *P3 does not outperform some baselines on high-quality datasets.*

In the end of Section 5, we explained why P3 shows more advantages  on low/medium data than on high-quality data. **In summary, as reflected by Fig. 4, exploring the whole Pareto front (P3) is important in searching a near-optimal policy when the data quality is low/medium, while carefully tuning the trade-off weight between the two objectives (other baselines) suffices to find a good policy if the data quality is high.** Hence, we observe less advantages of P3 on the high-quality datasets.

The reason behind is that the MDP models are very confident on high-return (realistic) state-action pairs if most samples in the training data are with high-return (high-quality), while they can be uncertain about many high-return pairs if the training data only cover a few high-return samples (low/medium quality). Hence, only the latter case requires exploring the whole Pareto front, where a near-optimal policy can reside in either the high-uncertainty + high return region or the low-uncertainty +  low return regions.

**P3 has more advantages in practice** because collecting high-quality datasets is usually expensive or infeasible in many real applications, which lack sufficient high-quality data. In these imperfect but practical scenarios, P3 performs significantly better and more stably than existing model-based offline RL methods that only learns one single policy.

### **Q3** (Reviewer EL3w and ngyt) *Exponential and temperature applied to the uncertainty in Eq. (4).*

**Uncertainty $u$ is non-negative but is not bounded**. We apply exponential to it so its value lies in the same range $[0,1]$ as the model return $r$. This normalization facilitates the tuning of their trade-off weight in Eq. (4) and also simplifies the generation/selection of reference vectors in Eq. (5).

**Tuning the temperature is necessary to encourage the smoothness of uncertainty across steps.** It helps to avoid one step’s uncertainty dominating the uncertainty of the whole trajectory. We conducted a grid search between $[1,3]$ and selected $\kappa=1.5$ for all benchmark datasets.

---

### Author Response · Authors · 2021-11-23
**Summary of Changes in the New Version**

We appreciate all reviewers for their insightful and constructive comments! We carefully addressed all the raised concerns and accordingly modified the paper. We highlight the modified parts with blue color in the latest version and here is a summary of the new changes for your convenience:

- **[New experiments]** We added a **new variant of P3 replacing online policy evaluation with an offline policy evaluation method “FQE”** for Pareto policy selection during inference. In this way, P3+FQE requires the same inference cost as other baselines. The details of FQE’s implementation and algorithm are provided in Appendix A.4. The experimental results of P3 with FQE inference are reported in Table 1, in which “P3+FQE” still outperforms all baselines on all low/medium-quality datasets. Moreover, we showed a strong correlation between FQE and online policy evaluation results in Appendix A.7, which indicates that offline policy evaluation enables more efficient inference for P3.
- We added **detailed discussions about the hyperparameter settings of the baselines** in Section 5.
- At the end of Section 5, we explained why **P3 shows more advantages on low/medium data than on high-quality data.** We also argued that collecting high-quality datasets is usually expensive or infeasible in practice, so P3 has more advantages in these imperfect but practical scenarios.
- We moved the Related Work section to the main paper and **discussed model-free offline RL methods such as DiME and their major difference to P3**, as suggested by Reviewer EL3w.
- We moved the Conclusion section to the main paper.
- We improved the writing of the last paragraph in Section 4.1.
- We fixed all typos and grammatical errors in the paper.

---

### Decision · Program_Chairs · 2022-01-20

**Decision:**

Accept (Poster)

**Comment:**

Previous approaches to model-based offline RL require carefully tuning the trade-off between model return and uncertainty. The authors propose an approach that produces a diverse pool of policies on the Pareto front of this tradeoff. On the D4RL offline RL benchmark, P3 outperforms competing approaches when the experience is collected with low or medium return policies.

Before the rebuttal, reviewers identified the following primary concerns:
* The experimental evaluation of P3 uses many policy evaluations to select the policy, which results in an unfair comparison with existing methods.
* P3 underperforms existing methods on some datasets. Why?

Overall, reviewers were satisfied by the response and raised their scores accordingly. The authors responded by including a modification of P3 that uses FQE to select the policy for evaluation, resolving the first concern. The authors explain that P3 underperforms on high return datasets because it splits its updates across the pool of policies. The authors state, "We believe (and in theory it does hold) that P3 can achieve the same performance of UWAC on high-quality datasets, if provided with more computational budget." I suggest that the authors conduct at least one experiment to verify this claim.

The proposed idea is interesting and the revisions the authors have made resolved the primary concerns from reviewers, so I recommend acceptance. The reviewer/author discussion has many substantial points that I recommend the authors integrate into the revision.